# Estimating real driving emissions from MAX-DOAS measurements at the A60 motorway near Mainz, Germany

Bianca Lauster[1], Steffen Dörner[1], Steffen Beirle[1], Sebastian Donner[1], Sergey Gromov[1], Katharina Uhlmannsiek[1], and Thomas Wagner[1]

[1]Max Planck Institute for Chemistry, Mainz, Germany

**Correspondence:** B. Lauster (b.lauster@mpic.de)

**Abstract.** In urban areas, road traffic is a dominant source of nitrogen oxides ($NO_x = NO + NO_2$). Although the emissions from individual vehicles are regulated by the European emission standards, real driving emissions often exceed these limits. In this study, two MAX-DOAS instruments on opposite sides of the motorway were used to measure the $NO_2$ absorption caused by road traffic at the A60 motorway close to Mainz, Germany. In combination with wind data, the total $NO_x$ emissions for the occurring traffic volume can be estimated. Hereto, the ozone-dependent photochemical equilibrium between NO and $NO_2$ is considered. We show that for 10 May 2019 the measured emissions exceed the maximum expected emissions calculated from the European emission standards for standardised test cycles by a factor of $11 \pm 7$. One major advantage of the method used here is that MAX-DOAS measurements are very sensitive to the integrated $NO_2$ concentration close to the surface. Thus, all emitted $NO_2$ molecules are detected independently from their altitude and therefore the whole emission plume originating from the nearby motorway is captured which is a key advantage compared to other approaches such as in-situ measurements.

## 1 Introduction

Nitrogen oxides ($NO_x$) is a collective term for nitrogen dioxide ($NO_2$) and nitric oxide (NO). In the troposphere, a photochemical reaction with ozone leads to an equilibrium state between $NO_2$ and NO (Pandis and Seinfeld, 2006). About three-quarters of the global emissions of $NO_x$ originate from anthropogenic sources (IPCC, 2013). Moreover, nitrogen oxides do not only play a major role in atmospheric chemistry but are also important in terms of air quality. The World Health Organization reports negative short-term as well as long-term exposure effects in pulmonary function and in other organs (World Health Organization et al., 2000). For this reason, the limitation of the concentration of nitrogen oxides is part of the European programme regarding ambient air quality and cleaner air (European Parliament and Council of the European Union, 2008).

Fossil fuel combustion from road traffic is a major contributor to $NO_x$ emissions. Hence, the European emission standards were introduced to regulate the exhaust emissions of new vehicles in the EU since 1998 (European Parliament and Council of the European Union, 1998) and tightened in 2007 by a new regulation bringing into force the so-called Euro 5 and Euro 6 norms (European Parliament and Council of the European Union, 2007) for passenger cars. New vehicles sold in the EU need to undergo a type-approval procedure which verifies the compliance with these regulations. This procedure is standardised depending on the emission class, e.g. by the New European Driving Cycle (NEDC; European Parliament and Council of the

European Union, 1970) and since 2017 by the Worldwide harmonized Light vehicles Test Procedure (WLTP; Council of the European Union, 2017). These include the measurement of exhaust emissions on a chassis dynamometer. Similarly, there are standards applying to heavy duty vehicles.

However, various studies (Carslaw et al., 2011; Chen and Borken-Kleefeld, 2014) have shown that the real driving conditions are more dynamic than the tested driving cycles. In addition, it is known that several manufacturers have installed software
that manipulates the test results by reducing emissions specifically during the test procedure (Borgeest, 2017). This results in increased exhaust emissions during normal driving operation.

In-situ measurements such as used in vehicle chasing experiments, e.g. performed by Pöhler and Engel (2019), directly measure the exhaust plume of individual vehicles. Others use remote sensing techniques (Carslaw et al., 2011; Chen and Borken-Kleefeld, 2014) to measure exhaust gases across-road. Both approaches are able to resolve the emission of individual
vehicles but it is difficult to derive representative fleet average emission factors, e.g. to compare these with expected emissions from models, as large data sets would be required.

Nevertheless, in the atmosphere NO and $NO_2$ form an equilibrium state which is mainly influenced by the ozone concentration and solar irradiance but not the primary composition and amount of the exhaust gases. Thus, the Multi AXis Differential Optical Absorption Spectroscopy (MAX-DOAS) yields a key advantage when operated at some distance from the emission
source. This method is described in more detail in the next section. The presented results are based on one day of measurements (10 May 2019) for proof of concepts. Further measurements could then be used to analyse, e.g., different driving conditions in more detail.

## 2 Method

The MAX-DOAS method (Platt and Stutz, 2008) allows measuring the differential slant column density (DSCD) of different
trace gases (Hönninger et al., 2004). Hereto, spectra of scattered sunlight are recorded at different elevation angles using ground based instruments. To convert the slant column density (SCD), which represents the integrated concentration along the slant light path, into the vertical column density (VCD), the so-called air mass factor (AMF) is needed. For trace gas layers close to the ground the geometric approximation for the AMF can be used (Hönninger et al., 2004). The integrated trace gas concentration along the vertical path is then given by

$$\text{VCD} = \frac{\text{SCD}}{\text{AMF}} \approx \sin(\alpha) \cdot \text{SCD} \tag{1}$$

where $\alpha$ is the elevation angle.

In order to remove the Fraunhofer lines, the logarithm of a so-called Fraunhofer reference spectrum with preferably minimal trace gas absorption is subtracted from the logarithm of the measured spectra. To fulfil this criterion, the reference spectrum is usually recorded with an elevation angle $\alpha = 90°$, i.e. in zenith direction. It can be assumed that for a given solar zenith angle
the stratospheric absorption is constant for measurements at different elevation angles. Then, the differential SCD yields the integrated tropospheric concentration of a specific trace gas along the light path for an altitude range from the surface up to about 2 to 3 km (Frieß et al., 2019, and references therein), i.e. the column density relative to the reference spectrum.

In this study, the MAX-DOAS method is used to quantify the $NO_x$ emissions of vehicles on a motorway. Using two MAX-DOAS instruments on the two sides of the motorway allows to measure the background $NO_2$ DSCDs on the upwind side and additionally the traffic induced $NO_2$ on the downwind side. The background $NO_2$ DSCD is then subtracted from the $NO_2$ DSCD on the downwind side and thus yields the $NO_2$ SCD caused by the traffic emissions. In a final step, the derived $NO_2$ SCD is converted into $NO_x$ emissions by combining it with wind data and assuming a steady state $NO_x$ to $NO_2$ ratio. These steps are described in detail below.

## 2.1 Experimental setup

To retrieve the amount of $NO_x$ emitted by road traffic, two Tube MAX-DOAS instruments (Donner, 2016) were set up on each side of a motorway. With these instruments, it is possible to measure the $NO_2$ DSCD of the ambient air along the viewing direction. The chosen measurement site is located along the heavily used A60 motorway close to Mainz, Germany, and has a long straight section which provides an advantageous geometry for the measurement setup. The exact alignment of the instruments for the presented measurement day is depicted in Fig. 1 and shows that the viewing direction is northward and parallel to the lane of traffic.

The chosen motorway section has a speed limit of $100 \, \mathrm{km \, h^{-1}}$. The next access and exit is about $1 \, \mathrm{km}$ in one direction and $1.5 \, \mathrm{km}$ in the other direction. Acceleration and deceleration should, therefore, only have a minor effect at the measurement site.

On the measurement day, continuous westerly wind was present so that the air mass transport was almost perpendicular to the motorway as well as to the viewing direction of the instruments. From the difference between the upwind (Instrument A, west side) and downwind (Instrument B, east side) signals, the emissions of the motorway are estimated. The locations of the instruments were about $160 \, \mathrm{m}$ and $220 \, \mathrm{m}$ to the west and east side of the motorway, respectively. Therefore, the area enclosed by the two Tube MAX-DOAS instruments contains the motorway section and a railway track. Possible sources of $NO_x$ are thus traffic emissions from cars, trucks and trains since no other sources (e.g. fires) were detected in the area.

Measurements were taken at an elevation angle of $20°$ and with a total integration time of $2 \, \mathrm{s}$. The short integration time favours a high temporal resolution even if the quality of the spectral fit (Sect. 2.2) decreases slightly at the same time. Assuming a plume height of up to $200 \, \mathrm{m}$, the area of highest sensitivity can be estimated. It is also depicted in Fig. 1 by the green shading. The choice of a rather high elevation angle constrains not only the sensitivity region but also decreases the influence of variations in the background signal by reducing the light path length in the lowermost atmosphere. It should be noted that there were broken clouds on the measurement day which possibly induce differences between the two instruments. This effect is further analysed in Sect. 2.3.

In addition, a camera and a weather station were positioned on the upwind side to obtain further information. Taking videos with this setup makes it possible to observe the traffic density on the motorway. The weather station records the wind direction and wind velocity as well as several other meteorological parameters such as pressure and temperature every second.

 ## 2.2   Spectral analysis

The spectral analysis of the obtained spectra is performed using the QDOAS software (version 2.112.2, Danckaert et al., 2012). As a reference, a series of $90°$ measurements were taken simultaneously with both Tube MAX-DOAS instruments at the upwind measurement site. In order to categorise differences between the two instruments (Sect. 2.3), reference measurements were taken at the same location after the measurement series was completed on both sides of the motorway. The wavelength calibration is accomplished using a high resolution solar spectrum (Chance and Kurucz, 2010). For the analysis, a wavelength range of $400\,\mathrm{nm}$ to $460\,\mathrm{nm}$ was selected. The DOAS fit settings are summarised in Table 1. The spectral analysis is run separately for each instrument yielding the $NO_2$ DSCD time series for both measurement sites. Exemplary, the QDOAS fit for one spectrum (Instrument A, upwind) is depicted in Fig. A1.

## 2.3   Instrumental differences

To estimate the influence of instrumental differences between the two Tube MAX-DOAS instruments on the $NO_2$ results, the reference spectra are investigated in more detail. These measurements were taken simultaneously with both instruments on the upwind side with an elevation angle of $90°$ (zenith view). Fig. 2 shows the time series of the $NO_2$ results for these spectra. The first $90°$ measurement of each instrument is taken as a reference.

As can be seen, in the grey-shaded area the standard deviation of the difference of the two instruments only amounts to

$$\Delta(NO_2\ \mathrm{DSCD}) = 0.4 \times 10^{14}\,\mathrm{molec\,cm^{-2}}. \tag{2}$$

Hereto, the data points of instrument B were interpolated to the time axis of instrument A. For the spectra after 15:05 UTC the signal differs widely where the difference between both instruments is characterised by a standard deviation of $7.9 \times 10^{14}\,\mathrm{molec\,cm^{-2}}$. This increased deviation is due to clouds passing by (see Sect. B1). Thus, the measurements in the grey-shaded area show that both instruments measure similar $NO_2$ DSCDs for the same measurement conditions, i.e. the same setup, viewing direction and cloud conditions. Therefore, these spectra are being accumulated to minimise noise and used as fixed references which assures that both instruments are analysed under the same conditions. The offset between the two instruments, which is visible in Fig. 2, originates from instrument properties and is constant in time. Thus, choosing the same reference period also compensates for this difference in the further processing of the data.

## 2.4   Integration time

In order to investigate the influence of the integration time on the spectral analysis, the fitting procedure is performed for spectra with different integration times but the same fit settings. Therefore, two or more spectra are added before performing the DOAS fit. The result of the $NO_2$ retrieval as well as the fit error and the average root mean square (RMS) over each measurement series is depicted in Fig. 3. As expected, the $NO_2$ fit error, which is given by the QDOAS analysis, shows the same trend as the RMS. For the short integration time of our measurements, the spectral residual of the fit is dominated by photon shot noise. This is also clearly demonstrated by the observed dependence of the RMS (and the fit error) on integration

time. The RMS decreases for longer integration times as the ratio of the measured signal to the photon shot noise increases. In contrast to the fit error decreasing with integration time, the $NO_2$ retrieval yields the same average $NO_2$ DSCDs for different integration times. Nonetheless, for longer integration times with decreased noise levels (as expected) also systematic structures appear in the residuals for one of the instruments. Detailed analyses showed that they are almost identical for all spectra and there is no systematic evolution with time. This is also consistent with the fact that there is no change in the result of the $NO_2$ retrieval regardless of the integration time. Therefore, it can be concluded that these systematic errors are unproblematic for this setup and analysis procedure. The standard deviation between the results for different integration times amounts to less than $0.2 \times 10^{14}$ molec cm$^{-2}$ for the east side and $0.7 \times 10^{14}$ molec cm$^{-2}$ for the west side measurements which is two orders of magnitude smaller than the $NO_2$ signal. Consequently, the measurements taken with an integration time of $2\,\mathrm{s}$ give sufficient results above the detection limit. This is preferable as high temporal resolution is necessary to resolve specific traffic events. In the following we focus on time-averaged emissions, as for the presented measurement day no individual emission plumes could be identified. Nevertheless, it is conceivable that a detection is possible for lower traffic volume (e.g. on Sundays) or higher workload (e.g. motorway sections with higher slopes).

## 3 Results

### 3.1 Measurement results

The measurement results for 10 May 2019 are shown in Fig. 4. Panel (A) depicts the time series of the measured $NO_2$ DSCDs for both, the upwind and downwind side, analysed as described in Sect. 2.2. The next panel (B) shows the difference between both signals

$$\mathrm{SCD}_{\mathrm{traffic}} = \mathrm{DSCD}_{\mathrm{downwind}} - \mathrm{DSCD}_{\mathrm{upwind}}. \tag{3}$$

A persistent offset is found with a mean value of

$$\overline{\mathrm{SCD}}_{\mathrm{traffic}} = (0.18 \pm 0.04) \times 10^{16}\,\mathrm{molec\,cm^{-2}} \tag{4}$$

as represented by the orange line. The error is calculated using error propagation of the standard errors of the mean for both instruments and additionally includes the deviation $\Delta(NO_2$ DSCD) between both instruments as derived in Sect. 2.3. Moreover, an uncertainty of $16\,\%$ is added to account for the impact of the broken clouds on the measurement time series. In order to investigate the effect, a cloud filter is applied as discussed in Sect. B2.

As there are no large sources of $NO_2$ other than the motorway close to the measurement site, the background $NO_2$ DSCDs in both measurements can be assumed to be the same. Therefore, the difference between both sides is most likely due to traffic emissions. There seems to be no significant additional emission due to the passing trains (marked by the dashed grey lines in the Figure) although the railway next to the measurement site is only used by diesel trains. Temporal variations can be found in the derived difference in addition to the constant offset. Panel (C) depicts the amount of traffic observed during the measurement period for which the number of vehicles was counted over one-minute intervals on a sample basis using

the recorded videos. (D) presents the wind direction measured by the weather station at the upwind side. (E) shows the wind velocity $v_{\text{wind},\perp}$ perpendicular to the viewing direction. The calculation is detailed in the next section.

## 3.2 Plume age

For a better understanding of the retrieved signal, the wind field needs further investigation. The quantity of interest is the wind velocity $v_{\text{wind},\perp}$ perpendicular to the viewing direction of the Tube MAX-DOAS instruments whose viewing directions are assumed to be parallel to the motorway. Thereby, the age of the measured plume can be quantified which is needed to retrieve the total emission (Sect. 3.3). The perpendicular wind velocity $v_{\text{wind},\perp}$ is shown in Fig. 4 (E) and is calculated using the wind velocity and direction as measured by the weather station. From the alignment of the two Tube MAX-DOAS instruments as

depicted in Fig. 1, it can be seen that the viewing direction corresponds to approx. $330°$. The perpendicular wind velocity is thus

$$v_{\text{wind},\perp} = v_{\text{wind,meas}} \cdot \cos(\phi_{\text{wind}}) \tag{5}$$

with

$$\phi_{\text{wind}} = \phi_{\text{wind,meas}} - 330° + 90°, \tag{6}$$

where $\phi_{\text{wind,meas}}$ is the measured wind direction at the weather station. The error can be calculated using the propagation of uncertainty principle. Hereto, the error of the wind velocity is estimated using the minimum and maximum values over $1\,\text{s}$ (with a sampling rate of $4\,\text{Hz}$). An additional error for a possible misalignment of the weather station with regard to the viewing direction of the telescopes of $2°$ is considered. During the measurement period, the wind velocity perpendicular to the viewing direction is at ground level on average

$$\overline{v}_{\text{wind},\perp} = (2.8 \pm 1.0)\,\text{m}\,\text{s}^{-1}. \tag{7}$$

Effects such as turbulence, especially in the vicinity of the motorway, and changing wind fields at plume height lead to uncertainties which can, however, not be readily quantified.

Taking into account the average distance between the motorway and the downwind instrument's viewing direction $x = (195 \pm 25)\,\text{m}$ estimated from Fig. 1 within the main area of high sensitivity, an average age of an air parcel of

$$t = (1.2 \pm 0.4)\,\text{min} \tag{8}$$

can be obtained. However, variations in the wind velocity and wind direction on short time scales affect the transport of an air parcel. Therefore, the plume age cannot always be correctly represented by Eq. 8. The correlation between the wind field and the measured $NO_2$ SCDs is further discussed in Sect. B3. Concluding, a constant wind velocity is favourable when applying this method.

## 3.3 Estimation of real driving emissions

To estimate the real driving emissions, first the mean $NO_2$ SCD must be converted into a VCD using the geometric approximation as given in Eq. 1. Thus, for the elevation angle of $(20 \pm 2)°$, the AMF amounts to $2.9 \pm 0.3$ and the measurement yields

$$\overline{VCD}_{traffic} = (0.6 \pm 0.1) \times 10^{19} \, \mathrm{molec \, m^{-2}}. \tag{9}$$

Multiplying this value by the average wind velocity perpendicular to the viewing direction, the measured emission of $NO_2$ amounts to

$$E_{meas, \, NO_2} = (1.8 \pm 0.7) \times 10^{19} \, \mathrm{molec \, (m \, s)^{-1}}. \tag{10}$$

This value now describes the number of molecules emitted per meter and second along the motorway section. It is a direct quantity of the measurements and can be converted into emissions per vehicle per second by dividing by the number of vehicles per length of the motorway.

In combustion processes, $N_2$ is mainly oxidised into NO and in the atmosphere it is further oxidised into $NO_2$ and other oxides of nitrogen (Pandis and Seinfeld, 2006) forming an equilibrium between NO and $NO_2$. Especially diesel vehicles also directly emit $NO_2$ (Carslaw et al., 2011, and references therein). Therefore, to retrieve the total $NO_x$ emissions from the observed $NO_2$ levels, the share of $NO_2$ in total $NO_x$ at the measurement site has to be known.

In order to estimate the rate of NO to $NO_2$ conversion, we used the CAABA box-model simulation with representative environment conditions and a road traffic source for the measurement period. CAABA uses the atmospheric chemistry model MECCA that includes the state of the art chemical mechanisms (Sander et al., 2019). A fraction of the traffic-emitted NO is photochemically equilibrated with air $NO_2$ at the daytime near-surface conditions. Hereto, the solar radiation is calculated for clear-sky using the solar inclination at the measurement location.

One important factor regarding the conversion is the ambient ozone level, as it regulates the photochemical $NO_x$ cycling and influences the resulting NO to $NO_2$ repartitioning dynamics. Where the emission fluxes are very high, the titration of ozone stops further conversion of NO to $NO_2$. However, turbulent mixing with ambient air increases with distance from the source and ozone in the air parcel containing the plume is replenished. Thereby, the conversion of NO to $NO_2$ continues. Our observations confirm that, for the presented measurement, ozone titration only prevails close to the emission source and thus has no significant influence on our measurements (see Sect. C1). Sufficiently high ambient ozone concentrations were measured at local environmental monitoring stations ($42 \, \mathrm{ppb}$ to $44 \, \mathrm{ppb}$, Mainz-Mombach, distance to the measurement site approx. $5 \, \mathrm{km}$, and Wiesbaden-Süd, approx. $9 \, \mathrm{km}$, Umweltbundesamt, 2019).

The corresponding $NO_x$ to $NO_2$ conversion factor, for the time $t = (1.2 \pm 0.4) \, \mathrm{min}$ an air parcel needs to get from the vehicle exhaust to the sensitivity region of the Tube MAX-DOAS instrument, can be deduced to be $f = 2.4 \pm 1.0$ (Sect. C1). The $NO_x$ emission is then derived using

$$E_{meas, \, NO_x} = f \cdot E_{meas, \, NO_2} \tag{11}$$

which equals

$$E_{\text{meas, NO}_x} = (4.3 \pm 2.5) \times 10^{19} \, \text{molec} \, (\text{m s})^{-1}. \tag{12}$$

In case the equilibrium is already reached, a conversion factor of $f_{\text{eq}} = 1.5$ needs to be applied instead. Then, the total $\text{NO}_x$ emission would amount to

$$E_{\text{meas, NO}_x, \text{eq}} = (2.7 \pm 1.1) \times 10^{19} \, \text{molec} \, (\text{m s})^{-1}. \tag{13}$$

The determination of the conversion factor $f$ relies on the rather rough estimate of the age of an air parcel as well as the ozone concentration and chemical processes during the measurement period. Therefore, the equilibrium value gives an estimate which is independent of these factors. However, it is rather unlikely that the equilibrium state is reached so close to the emission source (as also found for airborne measurements of emission fluxes from power plants; Meier, 2018). Nonetheless, the emission value $E_{\text{meas, NO}_x, \text{eq}}$ is within the error of $E_{\text{meas, NO}_x}$. In the following, the more realistic value of $E_{\text{meas, NO}_x}$ will be taken for the comparison with the expected traffic emissions.

### 3.4 Expected traffic emissions and comparison to real driving emissions

To calculate the expected traffic emissions, the emission per vehicle needs to be computed. The limiting values for $\text{NO}_x$ emissions, as given by the European emission standards, are summarised in Table 2. The limiting values for passenger cars are given in $\text{NO}_2$ equivalents per km depending on the fuel type. For trucks, the values are reported in $\text{NO}_2$ equivalents per kWh. To undertake the following calculation, the emission standards of trucks need to be converted into limiting values per km. Therefore, the values are multiplied by a conversion factor of $1.5 \pm 0.5 \, \text{kWh} \, \text{km}^{-1}$. This is composed of the fuel value $10.4 \, \text{kWh} \, \text{l}^{-1}$ of diesel fuel, the efficiency of a diesel engine of about $40 \, \%$ and an average consumption for trucks of $36 \, \text{l}$ per $100 \, \text{km}$ (Hilgers, 2016). The error accounts for varying fuel consumption of $\pm 10 \, \text{l}$ per $100 \, \text{km}$ and the uncertainty in the efficiency of the vehicle engine.

The European emission standards are theoretical values for the allowed emissions of different pollutants. They are, however, not the expected emissions under real driving conditions. In order to bring the values in line, so-called Real Driving Emissions (RDE) conformity factors are used for new emission norms (Euro 6d-temp; Council of the European Union, 2016). To avoid inconsistencies, in the following only the European emission standards serve to estimate the theoretically expected emissions.

For the calculations, the statistical composition of the vehicle fleet is considered (see Table 3). The passenger car fleet is broken down by registration districts, fuel types and emission groups. To analyse the emission per vehicle, the statistical distribution of Rheinhessen-Pfalz is chosen. This also includes the city of Mainz and the Mainz-Bingen region. Note that in this area more cars with old emission standards (Euro 3 and 4) are registered compared to the average in Germany. The relative number of trucks is broken down by emission group only and relates to the distance travelled by German trucks. Attention should be paid to the fact that non-German trucks account for about $35 \, \%$ of the total distance travelled in Germany (Kraftfahrt-Bundesamt, 2017).

From the emission standards and the statistical composition of the vehicle fleet, a theoretical emission per vehicle can be calculated. The weighted average of the emission limits amounts to

$$E_{\text{limit, cars}} = (116 \pm 5) \, \text{mg km}^{-1} \tag{14}$$

and

$$E_{\text{limit, trucks}} = (1248 \pm 277) \, \text{mg km}^{-1} \tag{15}$$

for passenger cars and trucks, respectively. The observed amount of traffic is deduced by counting the vehicles as shown in Fig. 4 and shows average values of

$$N_{\text{cars}} = (91 \pm 4) \, \text{min}^{-1} \tag{16}$$

and

$$N_{\text{trucks}} = (6 \pm 2) \, \text{min}^{-1}. \tag{17}$$

The error estimation accounts for miscounting the number of vehicles on the video e.g. when a truck shields the view of the other traffic lanes. Taking into account the average traffic volume, the theoretical total emission for the measuring period is given by

$$E_{\text{calc, NO}_x} = N_{\text{cars}} \cdot E_{\text{limit, cars}} + N_{\text{trucks}} \cdot E_{\text{limit, trucks}} \tag{18}$$

which yields

$$E_{\text{calc, NO}_x} = (0.4 \pm 0.1) \times 10^{19} \, \text{molec} \, (\text{m s})^{-1}. \tag{19}$$

Here, it is used that the $NO_x$ emissions are given in $NO_2$ equivalents. Thus considering the molar mass of $NO_2$ of $46 \, \text{g mol}^{-1}$ (Haynes, 2014), $1 \, \text{mg}$ of $NO_x$ emissions correspond to $1.3 \times 10^{19} \, \text{molec}$.

The theoretical emissions calculated from the European emission standards for standardised test cycles can now be compared to the measured $NO_x$ emissions. Evidently, for 10 May 2019 the measured amount of $NO_x$ is by a factor $11 \pm 7$ larger than theoretically expected. Even if an equilibrium state between $NO$ and $NO_2$ for the measured traffic emissions was assumed, the measured $NO_x$ emissions still show a higher value (by a factor of $7 \pm 3$) compared to the calculated emissions. Moreover, in the very unlikely case that the exhaust gases primarily consist of $NO_2$ and the measured $NO_2$ difference directly equals the $NO_x$ emissions, this discrepancy remains unexplained. Possible error sources in the measurement cannot completely explain these differences.

As the traffic volume was relatively constant throughout the measurement period, it is more likely that the statistics do not reflect the vehicle fleet well enough and/or a large part of the vehicles does not meet the emission standards. Here, it should be noted that the deviations of the actual vehicle composition from the assumed one cannot be the sole reason for this factor.

Assuming that only Euro 3 diesel cars and Euro III trucks, i.e. the technical status quo of the year 2000, were driving during the measurement period, the expected traffic emission would amount to

$$E_{\text{calc, NO}_x\text{, Euro3/III}} = (2.0 \pm 0.5) \times 10^{19}\,\text{molec}\,(\text{m\,s})^{-1} \tag{20}$$

which is still lower than the measured emission. As today only a minor fraction of all vehicles is registered as Euro 3 cars and Euro III trucks, this worst case scenario is highly unlikely. Considering that especially non-German trucks more often drive with defective exhaust gas systems, these could lead to large emissions even exceeding the Euro III norm. Thereby, the discrepancy between the theoretical and measured emissions could be partly explained. However, trucks only account for parts of the total traffic volume. This again implies an excess of the European emission standards regarding $NO_x$ emissions also for a significant number of passenger cars.

## 3.5 Comparison to the HBEFA database

The Handbook Emission Factors for Road Transport (HBEFA, version 4.1, Notter et al., 2019) provides emission factors for all current vehicle categories as weighted average values for Germany. To draw a comparison to the results deduced in the previous sections, the vehicle categories "passenger cars" and "heavy duty vehicles" are used as these can be readily identified in the camera recordings of the motorway section. The aggregated emission factors for $NO_x$ especially show higher emissions of passenger cars as compared to the theoretical emission limits (see Sect. C2). This results in an average emission flux of

$$E_{\text{HBEFA, NO}_x} = (1.1 \pm 0.1) \times 10^{19}\,\text{molec}\,(\text{m\,s})^{-1} \tag{21}$$

which is roughly three times larger than expected from the European emission standards. Although the database provides modelled real driving emissions, there remains a discrepancy to the measurements of a factor $4 \pm 2$. In conclusion, our measurement method yields reasonable results and is able to quantify average emissions of the motorway section. Nonetheless, differences remain which cannot easily be attributed to a specific error source.

## 4 Conclusions

The measurement of $NO_x$ emissions at the A60 motorway close to Mainz, Germany, gives an estimate of real driving emissions. With two MAX-DOAS instruments set up on each side of a motorway, it is possible to retrieve the $NO_2$ signal caused by road traffic and calculate the total $NO_x$ emissions for the occurring traffic volume.

The most uncertain aspect during the analysis of the data was the age of the measured plume at the downwind side. It directly affects the conversion factor $f$ of the $NO_x$ to $NO_2$ ratio and thus the final result of the measured emission (Eq. 11). To further investigate the effect of the plume age, it is favourable to set up several MAX-DOAS instruments downwind with different distances to the motorway. Thereby, the setup of the instruments could be optimised and the equilibrium state of $NO_2$ for the given weather conditions can be measured. Hereto, a stable wind field is advantageous. This yields a more accurate conversion factor.

Other aspects such as the high ozone concentration and relatively constant wind are uncritical for the presented measurement day and allow to apply a constant conversion factor $f$ to the average emission. Although the changing cloud cover caused large fluctuations in the $NO_2$ DSCDs, filtering the data leads to only slightly lower emissions. Consequently, this effect cannot explain the difference between the measured and expected emissions.

The main possible error source regarding the derivation of the expected $NO_x$ emissions is the difference from the assumed vehicle fleet to the measured vehicle fleet. Although the statistics are relevant to the Mainz region, the exact composition remains unknown. However, the worst case calculation showed that the uncertainty of the vehicle fleet cannot explain the deviation from the measured emission. Presumably, a considerable amount of vehicles did not meet the European emission standards. Moreover, it must be assumed that a substantial number of trucks are non-German vehicles. Recent studies showed that a large fraction of these vehicles had conspicuously high emissions which indicate deactivated fuel cleaning units (Pöhler and Engel, 2019). These could also explain the temporal variations in the measured time series. Applying this method at different measurement sites, different driving conditions (e.g. the slope of the motorway section, the allowed speed limit, road works etc.) and the impact of the composition of the vehicle fleet could be investigated in more detail.

It can be concluded that the measured emissions on 10 May 2019 exceed the maximum expected emissions calculated from the European emission standards for standardised test cycles (Umweltbundesamt) by a factor of $11 \pm 7$. The comparison to the HBEFA database also indicates elevated emissions on that motorway section. This observation is in line with the work of other groups (Carslaw et al., 2011; Chen and Borken-Kleefeld, 2014; Pöhler and Engel, 2019). Especially, the whole plume originating from the nearby motorway was measured rather than individual vehicle plumes and hence the possibility that parts of the plume get overlooked can be neglected which is a key advantage compared to other approaches such as in-situ measurements.

*Data availability.* Measurement data are provided in the supplement.

**Appendix A**

**A1  QDOAS analysis**

This section exemplary includes a fit result (Fig. A1) of the QDOAS analysis for a spectrum of Instrument A (west side, upwind) at an elevation angle of $20°$ using $2\,s$ integration time. The fit settings are specified in Tab. 1.

## Appendix B

### B1 Effect of clouds on the reference spectra

Clouds can have a great impact on MAX-DOAS measurements. A change of light path is caused by the increased scattering probability in clouds as there are more particles compared to the ambient air. Furthermore, the wavelength dependency of the scattered light changes for particle scattering processes compared to pure Rayleigh scattering. This effect already occurs for aerosols and is even more pronounced for clouds.

There are different methods to identify and classify clouds. Here, the temporal variation of the colour index (Wagner et al., 2014) is used. The colour index (CI) is defined as the ratio of two radiance values at different wavelengths. In this case, the wavelengths $320\,\mathrm{nm}$ and $440\,\mathrm{nm}$ are chosen. Thereby, the wavelengths cover a large range to pronounce the effect of the wavelength dependency.

The CI is calculated for the $90°$ measurements (compare to Fig. 2) and the obtained temporal evolution is given in Fig. A2. An almost constant CI is expected for cloud free conditions in this time period. It can be seen that measurements after 15:05 UTC were affected by clouds. This leads to larger deviations in the retrieved $NO_2$ signal as shown in Sect. 2.3. The offset of the CI between the two instruments can be ascribed to the specific instrumental properties as the instruments are not absolutely radiometrically calibrated. Accordingly, the CI analysis also encourages the approach to use only $90°$ measurements in the grey-shaded area as a reference.

### B2 Effect of clouds on the measurement result

Calculating the CI as described in Sect. B1 for all spectra, a characteristic behaviour can be seen (Fig. A3). As high temporal variation indicates cloud cover, all spectra where the CI is below the reference $CI_{ref}$ are filtered. The reference was inferred by fitting a $2^{nd}$ order polynomial to the data and is depicted as dashed line. The filtered time series are displayed in Fig. A4. Recalculating the mean difference between the two measurement sites yields

$$\overline{SCD}_{\text{traffic, filtered}} = (0.156 \pm 0.009) \times 10^{16}\,\mathrm{molec\,cm^{-2}} \tag{B1}$$

which is about $16\,\%$ smaller compared to the unfiltered case.

### B3 Correlation to the wind field

Assuming a constant emission, the $NO_2$ difference is expected to be reciprocal to the wind velocity. However, an air parcel is also affected by obstacles such as trees and follows the turbulent flow of air. Furthermore, the wind varies on time scales of less than $1\,\mathrm{min}$ whereas the transport of the air parcel from the emission location to the sensitivity region of the MAX-DOAS instrument happens on larger time scales of $1\,\mathrm{min}$ or more. This means that the time of the wind measurement and the time of the $NO_2$ measurement are shifted by a time difference in which the wind might change strongly. Hence, the age of the air parcel cannot always be correctly represented by the simple calculation in Eq. 8.

To test this hypothesis, both - the wind measurements and the time series of the $NO_2$ differences - are averaged over a time period of $12\,\mathrm{min}$. Figure A5 shows the correlation between both quantities ($R^2 = 0.365$). The data points are fitted using the linear least squares method (LLS, orange line) as well as using the orthogonal distance regression (ODR, green line). Here, ODR is able to take into account the standard errors of the mean values in the fitting procedure (Cantrell, 2008). In doing so,

the slope of the fit increases and at the same time the intercept decreases. Comparing the fit results with the obtained emission $E_{\mathrm{meas,\ NO_2}}$ over the complete $NO_2$ measurement series as described in Sect. 3.3, a slope of about $5000 \pm 2000\,\mathrm{molec\,(m\,s)}^{-1}$ is expected. The fits from Figure A5 show slopes of $4230 \pm 208\,\mathrm{molec\,(m\,s)}^{-1}$ for the LLS and $7539 \pm 2013\,\mathrm{molec\,(m\,s)}^{-1}$ for the ODR method which are in agreement with the expected value.

Nevertheless, the weak correlation is not completely surprising because of the low variability of the wind velocity. Moreover,

a constant wind velocity is generally advantageous for the measurements.

**Appendix C**

**C1  CAABA-MECCA simulation**

Fig. A6 presents the results of the plume emission simulation using the CAABA-MECCA box-model (Sander et al., 2019). Applying representative environmental conditions for the measurement period, road emission was approximated with a $10\,\mathrm{s}$

emission pulse of NO into the box. From the evolution of changes of NO and $NO_2$ abundances in the air parcel with time, the $NO_2$ to $NO_x$ ratio of the plume is deduced.

To analyse the possible effect of ozone titration, a Gaussian dispersion model is applied. It uses Pasquill stability classes (Pandis and Seinfeld, 2006) based on the atmospheric stability of the measurement day. With this dispersion model, the extent of the emission plume is estimated and the $NO_2$ mixing ratio from our measurements is calculated. While turbulence induced

by the local topography and obstacles like trees is neglected, it helps to estimate the evolution of the $NO_2$ mixing ratio between emission source and measurement location. From the comparison of the dispersion model and the observations, it can be concluded that the ozone-poor chemical regime only prevails close to the emission source.

In order to consider this in the emission estimate calculation, the transport of the air parcel containing the plume is subdivided into two sections: 1) Close to the emission source we assume that only negligible amounts of NO are converted into $NO_2$ and

no further conversion takes place as ozone is depleted. 2) Turbulent mixing with ambient air refills the ozone reservoir and NO to $NO_2$ conversion can be described by the CAABA box-model simulations. For simplicity, the distance, which corresponds to the $NO_2$ mixing ratio matching the one simulated in the box-model, is chosen as the transition between both sections. Thereby, the time for NO to $NO_2$ conversion is shorter than without consideration of ozone limitations. The resulting $NO_x$ to $NO_2$ ratio of the measured air parcel is estimated to be $f = 2.4 \pm 1.0$.

## C2  The HBEFA database

The HBEFA (version 4.1, Notter et al., 2019) database provides emission factors for various common vehicle types (passenger cars, light and heavy duty vehicles, buses and coaches, and motorbikes). Here, they differentiate by emission standard (Euro 0 to Euro VI) and by different traffic situations. The database includes not only all regulated pollutants but also a number of non-regulated pollutants, including $CO_2$ and fuel/energy consumption.

In Sect. 3.5 the aggregated $NO_x$ emission factors are considered for the vehicle categories "passenger cars" and "heavy duty vehicles" regarding the year 2020. These yield values of

$$E_{\text{HBEFA, cars}} = 499\,\text{mg km}^{-1} \tag{C1}$$

and

$$E_{\text{HBEFA, trucks}} = 1426\,\text{mg km}^{-1} \tag{C2}$$

for the two vehicle categories, respectively. Applying the same calculations as in Sect. 3.4, i.e. taking into account the number of vehicles during the measuring period and using Eq. 18 with the emission values given by the HBEFA database, the emission flux can be estimated at

$$E_{\text{HBEFA, NO}_x} = (1.1 \pm 0.1) \times 10^{19}\,\text{molec}\,(\text{m s})^{-1}. \tag{C3}$$

*Author contributions.*  TW, SDÖ and BL designed the experiment. Adaptation of the instruments to the measurement setup was implemented by SDO, SDÖ and BL. SDÖ, BL and KU performed the measurements. SG developed and performed the simulations. BL prepared the manuscript with contributions from all co-authors. TW, SB, SDÖ and SDO contributed with valuable feedback and supervised the study.

*Competing interests.*  The authors declare that they have no conflicts of interests.

*Acknowledgements.*  We acknowledge the electronics workshop (i.e. Thomas Klimach, Christian Gurk, Mark Lamneck and Frank Helleis) and the mechanical workshop (i.e. Michael Dietrich and Ralf Wittkowski) of the Max-Planck Institute for Chemistry, Mainz, for the continuous support in the development of the Tube MAX-DOAS instrument. We are also thankful to Denis Pöhler (Airyx GmbH) for sharing his expertise on traffic emission estimation.

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

© Google Earth Pro: Mainz, 49°58'55.48"N 8°12'34.81"E, eye altitude 4.98 km.

Google 2018, GeoBasis-DE/BKG 2009., 2018.

Haynes, W. M.: CRC handbook of chemistry and physics, CRC press, 2014.

Hilgers, M.: Kraftstoffverbrauch und Verbrauchsoptimierung, Springer, https://doi.org/10.1007/978-3-658-12751-00, 2016.

Hönninger, G., Friedeburg, C. v., and Platt, U.: Multi axis differential optical absorption spectroscopy (MAX-DOAS), Atmospheric Chemistry and Physics, 4, 231–254, https://doi.org/10.5194/acp-4-231-20040, 2004.

IPCC: Climate Change 2013: The Physical Science Basis. Contribution of Working Group I to the Fifth Assessment Report of the Intergov-
ernmental Panel on Climate Change, Cambridge, UK and New York, New York, USA, 2013.

Kraftfahrt-Bundesamt: Lastfahrten im Inlandsverkehr nach Hauptverkehrsbeziehungen mit europäischen Lastkraftfahrzeugen im Jahr 2017, https://www.kba.de/DE/Statistik/Kraftverkehr/europaeischerLastkraftfahrzeuge/Inlandsverkehr/inlandsverkehr_node.html, last accessed: 18.07.2019, 2017.

Kraftfahrt-Bundesamt: FZ 1 Bestand an Kraftfahrzeugen und Kraftfahrzeuganhängern nach Zulassungsbezirken, 1. Januar 2019, https://
www.kba.de, last accessed: 26.02.2020, 2019a.

Kraftfahrt-Bundesamt: VD 5 Verkehr deutscher Lastkraftfahrzeuge Gesamtverkehr Mřz 2019, https://www.kba.de, last accessed: 19.02.2020, 2019b.

Kraus, S.: DOAS Intelligent System, institute of Environmental Physics, University of Heidelberg, Cooperation with Hoffmann Messtechnik GmbH, 2003.

Meier, A. C.: Measurements of horizontal trace gas distributions using airborne imaging differential optical absorption spectroscopy, Ph.D. thesis, Universität Bremen, 2018.

Notter, B., Keller, M., Althaus, H.-J., Cox, B., Knörr, W., Heidt, C., Biemann, K., Räder, D., and Jamet, M.: HBEFA (Handbook Emission Factors for Road Transport) 4.1 Development Report, https://www.hbefa.net/e/index.html, online version, Last accessed: 18.08.2020, 2019.

Pandis, S. N. and Seinfeld, J. H.: Atmospheric chemistry and physics: From air pollution to climate change, Wiley, 2006.

Platt, U. and Stutz, J.: Differential Optical Absorption Spectroscopy: Principles and Applications, Springer Science & Business Media, 2008.

Pöhler, D. and Engel, T.: Bestimmung von realen Lkw $NO_x$-Emissionen (Real Driving Emissions) und hohen Emittern auf deutschen Autobahnen, 2019.

Rothman, L., Gordon, I., Barber, R., Dothe, H., Gamache, R., Goldman, A., Perevalov, V., Tashkun, S., and Tennyson, J.: HITEMP,
the high-temperature molecular spectroscopic database, Journal of Quantitative Spectroscopy and Radiative Transfer, 111, 2139–2150, https://doi.org/10.1016/j.jqsrt.2010.05.001, 2010.

Sander, R., Baumgaertner, A., Cabrera-Perez, D., Frank, F., Gromov, S., Grooß, J.-U., Harder, H., Huijnen, V., Jöckel, P., Karydis, V. A., et al.: The community atmospheric chemistry box model CAABA/MECCA-4.0, Geoscientific model development, 12, 1365–1385, https://doi.org/10.5194/gmd-12-1365-20190, 2019.

Serdyuchenko, A., Gorshelev, V., Weber, M., Chehade, W., and Burrows, J.: High spectral resolution ozone absorption cross-sections–Part 2: Temperature dependence, Atmospheric Measurement Techniques, 7, 625–636, https://doi.org/10.5194/amt-7-625-20140, 2014.

Thalman, R. and Volkamer, R.: Temperature dependent absorption cross-sections of O 2–O 2 collision pairs between 340 and 630 nm and at atmospherically relevant pressure, Physical chemistry chemical physics, 15, 15 371–15 381, https://doi.org/10.1039/C3CP50968K, 2013.

Umweltbundesamt: Emissionsstandards, https://www.umweltbundesamt.de/themen/verkehr-laerm/emissionsstandards, last accessed:
480    06.07.2019.

Umweltbundesamt: Aktuelle Luftdaten, Fachgebiet II 4.2, Beurteilung der Luftqualität, https://www.umweltbundesamt.de/daten/luftbelastung/aktuelle-luftdaten, ozone concentration of 10 May 2019 at Mainz-Mombach (DERP007) and Wiesbaden-Süd (DEHE022), 2019.

Vandaele, A. C., Hermans, C., Simon, P. C., Carleer, M., Colin, R., Fally, S., Merienne, M.-F., Jenouvrier, A., and Coquart, B.: Measurements of the NO2 absorption cross-section from 42 000 cm- 1 to 10 000 cm- 1 (238–1000 nm) at 220 K and 294 K, Journal of Quantitative Spectroscopy and Radiative Transfer, 59, 171–184, 1998.

Wagner, T., Apituley, A., Beirle, S., Dörner, S., Friess, U., Remmers, J., and Shaiganfar, R.: Cloud detection and classification based on MAX-DOAS observations, Atmospheric Measurement Techniques, 7, 1289–1320, https://doi.org/10.5194/amt-7-1289-2014, 2014.

World Health Organization et al.: Air quality guidelines for Europe, 2000.

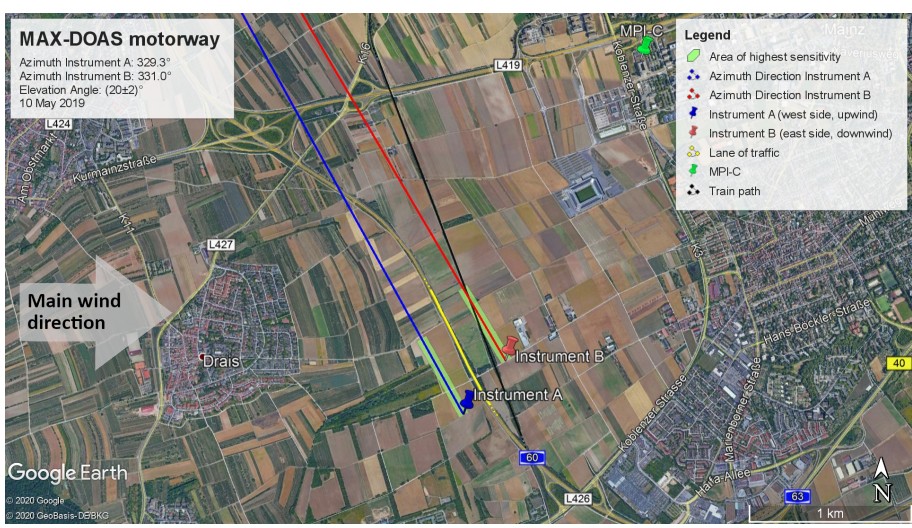

**Figure 1.** Alignment of the two Tube MAX-DOAS instruments on the measurement day, 10 May 2019. The instruments are located on both sides of the A60 motorway, Mainz, Germany, with a viewing direction parallel to the lane of traffic. The area between both instruments encloses the motorway and the railway track. Our measurements have highest sensitivity within the green-shaded area. On the measurement day, continuous wind from westerly directions was present. Created with © Google Earth Pro (2018).

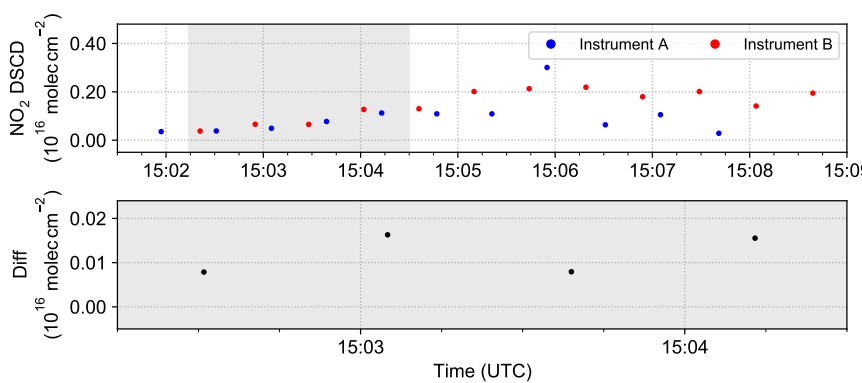

**Figure 2.** Time series of the $NO_2$ results for the 90° measurements of both instruments on the upwind side. The spectra are analysed using the first 90° spectrum as a reference. The grey-shaded area denotes the range where both measured similar $NO_2$ DSCDs. In the lower panel the difference between the two Tube MAX-DOAS instruments is depicted, zoomed into the grey-shaded area.

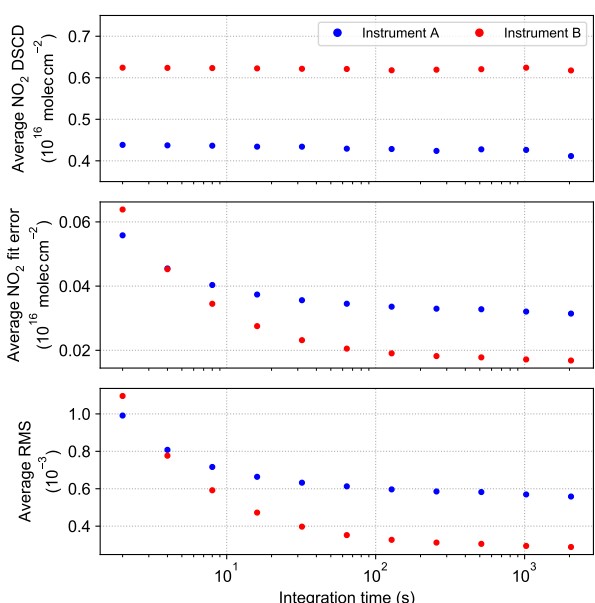

**Figure 3.** Average $NO_2$ DSCD, fit error and RMS for both measurement sites (Instrument A: west side, upwind; Instrument B: east side, downwind) for different integration times.

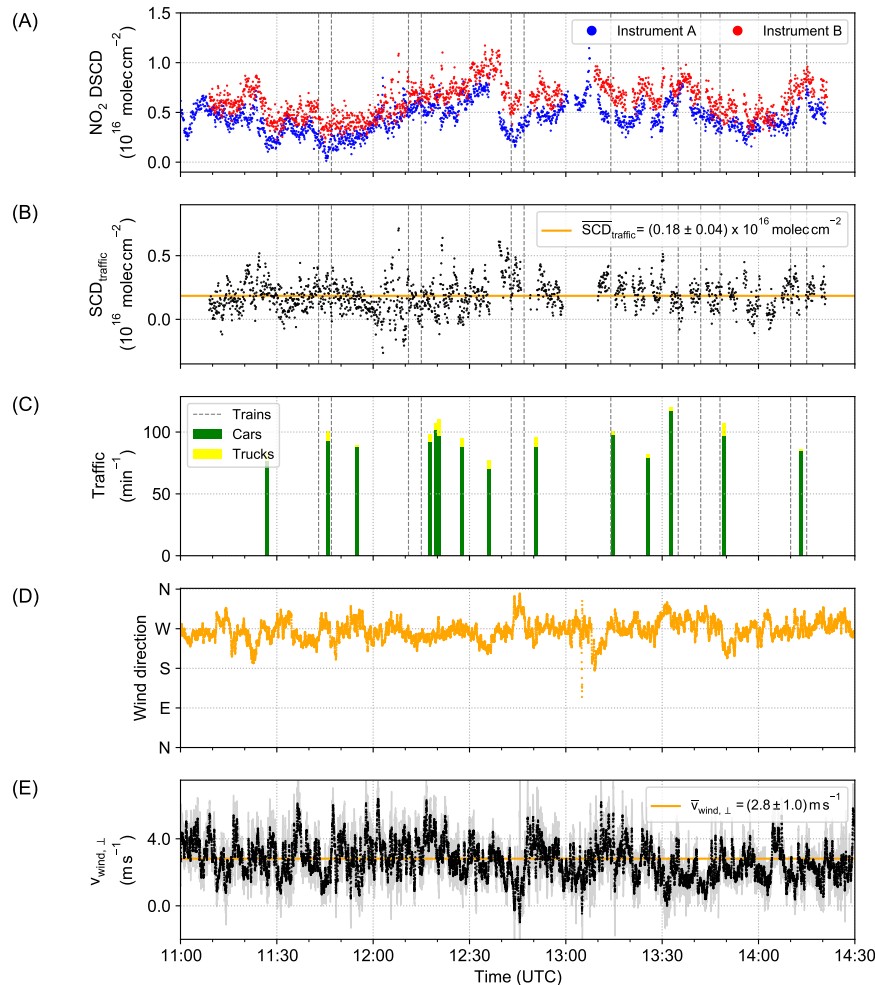

**Figure 4.** Time series of the measurement results of 10 May 2019. (A) depicts the measured $NO_2$ DSCD for both measurement sites (Instrument A: west side, upwind; Instrument B: east side, downwind). In (B) the difference $SCD_{traffic}$ between both signals is shown. The orange line symbolises the average value. (C) presents the traffic volume during the measuring duration. The number of vehicles was retrieved by counting from the videos over one-minute intervals on a sample basis. The dashed grey lines represent the times of passing trains. (D) depicts the wind direction as measured by the weather station. (E) shows the wind velocity $v_{wind,\perp}$ perpendicular to the viewing direction of the Tube MAX-DOAS instruments. The orange line denotes the mean value over the whole measurement period. Here, the light grey values depict the error $\Delta v_{wind,\perp}$ of the calculated wind velocity.

**Table 1.** Settings of the spectral analysis.

| Species | Cross-section |
|---|---|
| $NO_2$ | Vandaele et al. (1998), @ 298 K |
| $O_4$ | Thalman and Volkamer (2013), @ 293 K |
| $O_3$ | Serdyuchenko et al. (2014), @ 223 K |
| $H_2O$ | Rothman et al. (2010), HITEMP |
| Ring, Second Ring | Calculated with DOASIS (Kraus, 2003) using the reference spectrum |
| Polynomial | $5^{th}$ order |
| Offset | constant |

**Table 2.** European emission standards for $NO_x$ emissions (Umweltbundesamt).

For passenger cars separated into fuel types in $mg\,km^{-1}$ $NO_2$:

|        | Euro 3 | Euro 4 | Euro 5 | Euro 6 |
|--------|--------|--------|--------|--------|
| diesel | 500    | 250    | 180    | 80     |
| petrol | 150    | 80     | 60     | 60     |

For trucks in $mg\,kWh^{-1}$ $NO_2$:

| Euro III | Euro IV | Euro V | Euro VI |
|----------|---------|--------|---------|
| 5000     | 3500    | 2000   | 460     |

**Table 3.** Vehicle fleet composition by emission group in %.

For passenger cars (Kraftfahrt-Bundesamt, 2019a):

|        | Euro 3 | Euro 4 | Euro 5 | Euro 6 |
|--------|--------|--------|--------|--------|
| diesel | 3±1    | 6±1    | 11±1   | 8±1    |
| petrol | 6±1    | 23±1   | 16±1   | 16±1   |

For trucks (Kraftfahrt-Bundesamt, 2019b):

| Euro III | Euro IV | Euro V | Euro VI |
|----------|---------|--------|---------|
| 1±1      | 1±1     | 19±1   | 78±1    |

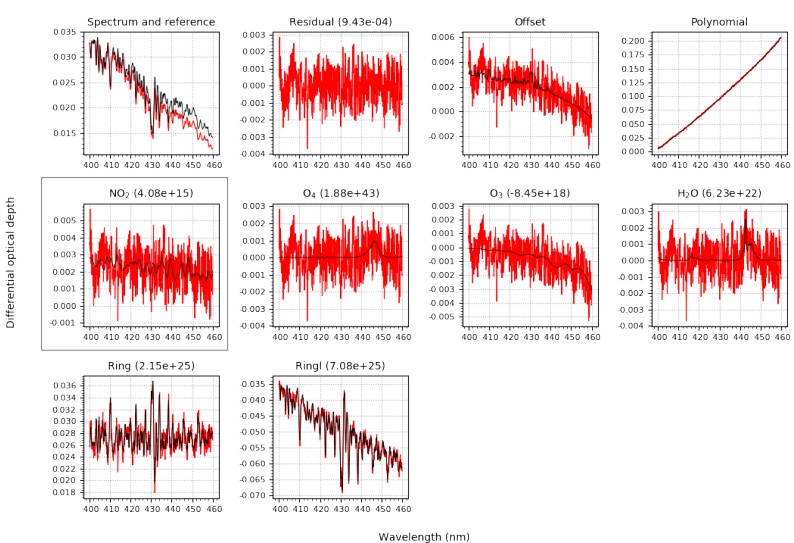

**Figure A1.** Example fit result of the QDOAS analysis (Instrument A: west side, upwind; 11:59:11 UTC, 10 May 2019). The measured optical densities of different absorbers are depicted in red whereas the fit results are depicted in black. The values in the titles refer to the resulting slant column densities in molec cm$^{-2}$. The error in the $NO_2$ fit amounts to $0.53 \times 10^{15}$ molec cm$^{-2}$.

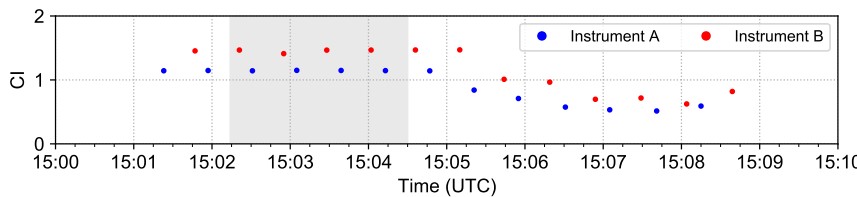

**Figure A2.** The temporal evolution of the colour index (CI, intensity ratio $320\,\mathrm{nm}/440\,\mathrm{nm}$) for the $90°$ measurements, which were taken simultaneously at the upwind side, is depicted. The grey-shaded area depicts the range where both instruments measured the same $NO_2$ signal (compare to Fig. 2).

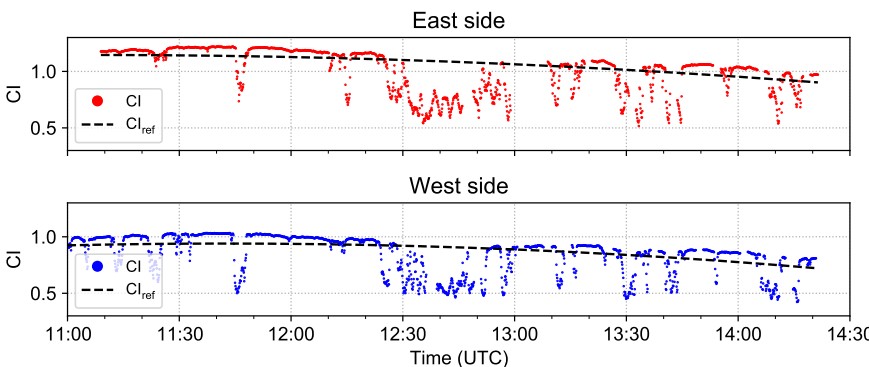

**Figure A3.** The colour index (CI, intensity ratio $320\,\mathrm{nm}/440\,\mathrm{nm}$) for both measurement series. The dashed line ($\mathrm{CI_{ref}}$) indicates the filter threshold.

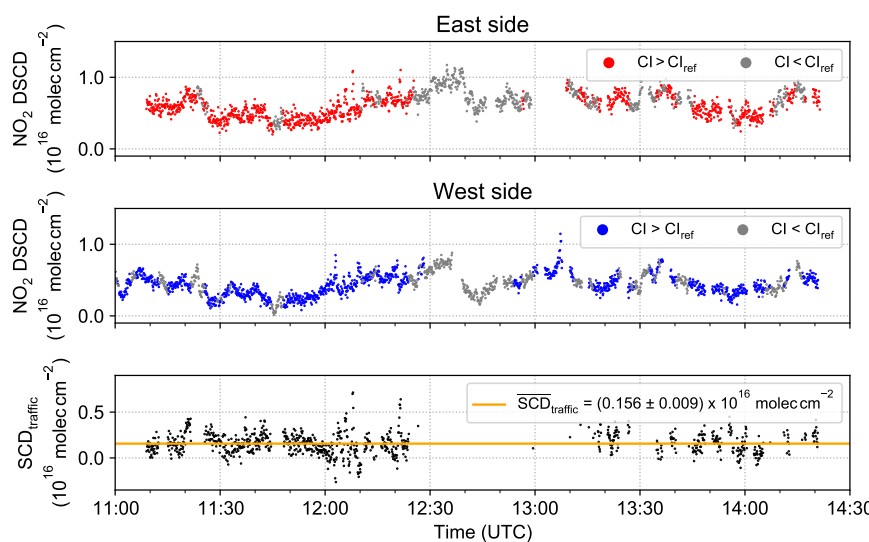

**Figure A4.** Analysis result of the $NO_2$ DSCDs for both sides (blue: west side, upwind; red: east side, downwind) with applied cloud filter based on the colour index (CI, intensity ratio $320\,nm/440\,nm$). The grey data points are filtered out. The resulting difference $SCD_{traffic}$ is depicted in the lowermost panel yielding slightly lower $NO_2$ SCDs compared to the unfiltered case.

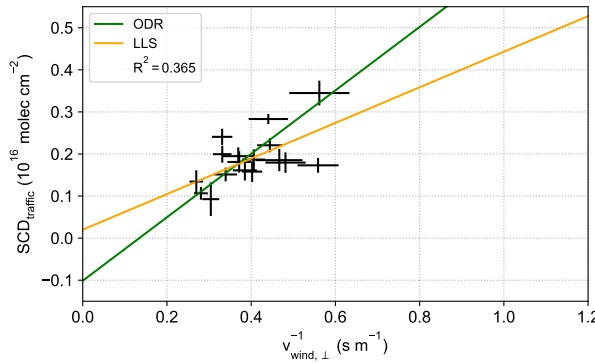

**Figure A5.** Correlation between the inverse of the wind velocity $v_{wind,\perp}^{-1}$ perpendicular to the viewing direction of the Tube MAX-DOAS instruments and the $NO_2$ signal ($SCD_{traffic}$) for a $12\,min$ averaging time span. The data points were fitted using the linear least squares method (LLS) and orthogonal distance regression (ODR).

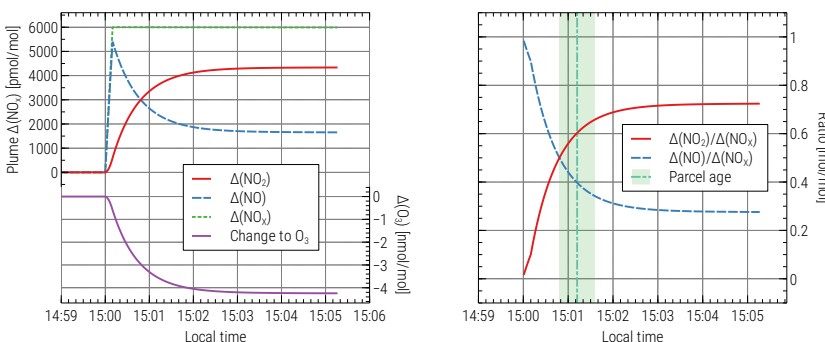

**Figure A6.** CAABA-MECCA box-model simulation for the presented measurement day using representative environmental conditions. At 15:00 simulated local time, a $10\,\mathrm{s}$ emission of NO into the box is performed representing the emission from road traffic. The left panel shows changes of $NO_x = NO + NO_2$ and ozone ($O_3$) compared to the background values. The right panel depicts the $NO_2$ to $NO_x$ as well as NO to $NO_x$ ratio in the plume.