# Peer review of "Estimating real driving emissions from MAX-DOAS measurements at the A60 motorway near Mainz, Germany"

_Atmospheric Measurement Techniques, 2020_

## Referee Comment (RC1) · Anonymous Referee #1 · 16 Jul 2020

The study by Lauster et al. describes a new method to quantify the NOx emissions from a motorway using two MAX-DOAS in parallel. This method is new and complementary to the existing ones, the analysis appears valid, and the presentation of the results in the paper is in general clear, although there is room for improvement in this respect. The experiment also addresses a hot topic regarding air quality. This work fits well in the scope of AMT. Therefore this work should be published, once the authors have taken into account the following remarks.

One limitation of this study is its small database. To my understanding, there was only one day of measurements (10 May 2019). This date appears in the main text only in

section 3 (I know it is in the caption of Fig. 1). It is fine to demonstrate a new technique with a small database, but this should be clear in the text. That means adding the date of experiment to the sentences of the abstract and conclusion which gives the factor 11+/-7. In case the authors performed more of such measurements but could only use those of May 10 for some reasons, it would be interesting to (briefly) explain what the problems were.

It s confusing that the legends indicate 'west side', 'east side' in Fig. 2 and Fig. A1, since they show measurements when both instruments were on the west side to record reference measurements. I suggest to label the instruments e.g. A and B across the text and figures instead (keeping the west side, east side where it makes sense).

l. 125 and below: Can the authors explain why they use the non filtered SCDtraffic estimate in the main text, if they have filtered the clouds in A2? Does the statement that the 'clouds have only a small impact' refer to the 16% of A2? If so, this is more important than the standard error of the mean (5%) and thus not 'a small impact'.

l.177: 'Our simulations with CAABA confirm...' -> The O3 concentration is indeed an important parameter is in the NO2/NO evaluation, one can imagine that the atmospheric mixing is as well. Could the authors add a figure with these simulations, e.g in the appendix? If the NO2/NO ratio is stable in the O3 conditions on 10 May 2019 in Mainz, it is interesting to know in which O3 conditions this ratio is not stable.

l. 188 The authors could refer to previous experiments which indicate that it is unlikely that the NO2/NO equilibrium would be reached so close to a source, e.g. the airborne measurements of NOx fluxes from power plants (see for instance the Phd of A. Meier, Uni. Bremen), or similar studies.

In appendix A1, the statement 'As for cloud free condition a constant CI is expected' is misleading since the CI varies, even without any clouds, with the sun position (see e.g. Gielen, 2014). In practice, this statement is only valid because of the short considered time period, please rephrase.

The last sentence of the appendix 'a constant wind is advantageous for the measurements' is an important take-home message and should be explicit in Sect 3.2 and in the conclusions.

I have several smaller suggestions to improve the presentation, see below.

Minor edits —

l.8: 'independent' -> independently ?

l.13: 'A large fraction of the global emissions' -> can the authors be quantitative on this fraction?

'Therefore'-> does not seem an appropriate adverb here since it s not linked by cause to the previous sentence, what about 'Moreover'?

l.30: 'need to convert NO into NO2 as they directly measure the exhaust plume'-> can the author briefly explain why? (the emissions are mainly NO?). It makes sense to detail also since the reference is in German.

l.55: 'the differential SCD yields the integrated tropo concentration of a specific trace gas'. This seems too short to be accurate. Please specify that the integration is along the photon path and that this quantity is relative (differential) to the column in the reference spectrum.

l.68: perpendicular -> almost perpendicular?

l. 71: 'Possible source of NOx' although that may seem obvious to the authors, I suggest to add that 'since no fire was detected in the area' for other readers

Spectral analysis: presenting the DOAS fit parameters (window, cross-sections, polynomial orders...) in a dedicated table would be more readable and synthetic.

l. 98: 'As can be seen in the grey area' -> actually not much can be seen in the grey area due to the y-axis scale of the lower subplot. I suggest to redo this figure 2, with

the second subplot zoomed in the time period of the grey area so that we really see that the delta is about 4e14. This would also make the subplots less redundant.

In the text, the authors should also explain what this delta is in practice (interpolated? one channel assumed constant?) since the measurements do not appear synchronized in time.

l. 104 'spectra are being integrated' -> '... averaged' ?

l. 285 'as shown in fig.2'-> 'fig. A2'?

l. 292 'in Fig. A3 where the dashed line indicates ... threshold' -> Fig A2 ?

l. 112 It s expected than the error trends follows the RMS, as it is expected that the RMS decreases with increasing integration times. Please add a few words on the physical explanation (shot noise ...)

l. 115-116 'Consequently ... to resolve specific traffic event' -> Please break this sentence in two for the sake of readability

l. 160 For the sake of readability, I suggest to be more explicit with the geometric approximation of the AMF at 20° i.e. to write 2.92.

Figure 4 is important and should be improved. The y axis of pannels A and B should be zoomed to better see the variations and mean values. Pannels E and F are redundant, the authors could only show one of them (leading to larger remaining subplots and a clearer figure).

l. 199 'These emission standards' -> 'the emission standards of trucks' (would be clearer for the reader)

---

## Referee Comment (RC2) · Anonymous Referee #2 · 21 Jul 2020

The manuscript present a new approach to derive average vehicle NOx emissions on a motorway using passive MAX-DOAS. This method is a further adaptation / modification of emission estimates like performed with car MAX-DOAS [e.g. Ibrahim et al. 2010] or stationary MAX-DOAS for volcanoes [Galle et al. 2010]. While the basic measurement principle is similar, the setup was here adapted to the task of vehicle emission measurements. The applied method are well described and clear. They have the potential to be established for general vehicle emission monitoring of whole fleets. The topic fits well in the scope of AMT. There are two major weak points. First, several parameters which influence the measurement are not well considered. They lead to a further increase of the derived emission factor error, which is already quite large with 4.3+/-

2.5 x 10ˆ19 molec/(ms). Second, the calculation of the expected NOx emissions are incorrect and underestimated, which make the manuscript and likely the measurement principle disputable. This does not mean that a significant higher emission is derived than expected, but a more realistic expected emission would strengthen the value of the manuscript. The manuscript should first be corrected before publication.

Mayor points.

Chapter 3.4: Expected traffic emissions The EURO emission standard is a limit which is based on a lab test cycle on a chassis dynamometer (NEFZ, and now WLTP) and only needs to be fulfilled on average over the whole test cycle. The emissions can be for some driving situations higher and for other lower. Especially on a motorway where the engine load is high, emissions are typically higher than the average especially for passenger cars (e.g. HBEFA data base, Lashkina and Lashkin 2016, Athanasios et al. 2019, for trucks e.g. TNO 2016). Second, It is expected and well known, that real driving emissions (RDE) will be higher as the driving and surrounding properties in the test cycle are not realistic (like also, mentioned in the manuscript l. 26.). For trucks this is also limited since EURO VI with a RDE factor of 1.5. This mean that EURO VI trucks are allowed to emit on average 1.5*460mg/kWh = 690mg/kWh in RDE (not the applied 460mg/kWh). For passenger cars the RDE confirmation factor is 2.1 since EURO-6d-Temp. RDE are thus for these diesel cars 2.1*80mg/km = 168mg/km. For older vehicles RDE is not tested and thus an emission confirmation with a confirmation factor is not defined. In conclusion this mean that the emission standard (Table 1) are not the expected RDE even if the vehicles confirm to the emission limit. The error become obvious as the expected weighted emission limit of the vehicle fleet (l. 213) is with 116mg/km below the RDE limit newest EURO 6d-temp diesel passenger cars need to confirm (168mg/km). Third, there are engine situations where significant higher emissions are allowed like cold start. If directly comparing measured emissions with calculated emissions, it should be excluded that these driving situations could contribute. Else they need to be considered.

There are different ways to handle the comparison more correctly: a) The expected emissions are modeled using the vehicle fleet, number, driving property at the measurement site and emission RDE data from HBEFA data base. This can also be made if it is expected that all vehicles confirm to the legislation or if included known RDE emission values. The expected emissions will increase in comparison to the authors calculation. That does not mean that they are than in agreement with the measurement, but this would allow a comparison between expected and measured average emissions.

b) The derived total emission is compared to the average emission limit on the chassis dynamometer (like currently done in the manuscript), however than a direct relation of how much the derived emissions are higher needs to be avoided. It must be clearly stated that the calculated emissions do not represent the expected emissions on the motorway, which is higher even if the vehicles confirm to the legislation. The comparison just gives the reader a relation between the numbers. In general the manuscript would than focus more on the derived total emission and less on the comparison.

c) The calculated expected emissions are at least more realistic. That mean that emission factors for motorways need to be used. Additional the RDE conformity factor need to be applied, which however only exist for newer EURO 6 / VI. How to deal with older cars is thus difficult. Additionally some estimated emissions of the trains need to be considered. Even if they are not clearly seen in the DSCD's (like mentioned in l. 130), they are still included in these data.

l. 209: The number of total traveled distance of trucks may not represent the real truck composition on the motorway. Especially on the motorway typically more foreign trucks are present than on average on the road. A more realistic number can be found from the toll collect system (https://www.bag.bund.de/DE/Navigation/Verkehrsaufgaben/Statistik/Mautstatistik/mautstatistik_node.html)

Chapter 2.2: Deriving DSCD's A spectral fit is missing in the appendix.

[Figure]

l. 110: The given NO2 DSCD error of 0.006 x 10ˆ16 molec cmˆ-2 does not agree to the given RMS values. Please provide the correct NO2 DSCD errors of the spectral analysis. The error for the average DSCD is not reducing with Gaussian error propagation. Also in l. 125 / 126 the given error seem to be calculated with gaussian error propagation of the mean which is not valid. Systematic measurement errors do not behave like a statistical standard error.

The typical averaging (2s) is very short with resulting noisy NO2 DSCD's. The authors derive an average emission factor over a longer time period. It is xplained that there is no signifficant difference between the different time resolutions (l. 113). It is not clear which difference the authors mean here. The one for the example in Fig. 3? Or the difference for the whole SCD_traffic? Even if the difference is small, I do not understand the argument in l. 116 "to resolve specific traffic events", as non of these events are analyzed in the manuscript. If analyzing averaged spectral data (16 s or longer) the section 2.4 and Fig. 3 can be shorten.

The influence of clouds is not clear through the manuscript. Clouds seem to have a large influence on the results. It is not clear if both MAX-DOAS point at 90° at the same location, why a difference in DSCD is observed? Both should see the same cloud and thus same variation of DSCD. From Fig. 2 it looks like a systematic offset for the "East side" instrument is observed. If such small variations already cause such large difference between the instrument (east side instrument measure higher DSCD), how can you exclude that this is not the case when the instrument measure at different locations at 20°? From Fig. 2 only the measurement situation without clouds are used for the reference. The argument in l. 132. that clouds have only a small influence is not clear as Fig. A3 show the influence also for the emission measurement. With the argument from Fig. 2 (both instrument at west side) also only data without clouds should be used for the emission analysis (like Fig. A3 instead data from Fig. 4).

Are the "West side" and "East side" instrument at the same height? If not, what would be the influence on the NO2 DSCD if they are not at the same height? Could this cause

some bias in the DSCD_traffic.

Chapter 2.2: Estimation of real driving emissions

l. 168: The vehicles emit also directly NO2. The share is for diesel engines (the main NOx emitters) rather high with 30%.

l. 170: Specify that "the share of NO2 in total NOx" need to be known "at the measurement location".

l. 174: The conversion of 2/3 of NO to NO2 is estimated to 4 minutes. This conversion needs O3. As NO emissions are very high at the emissions source, O3 is completely titrated, and thus NO can not further react to NO2. Even if the background conc. (l. 180) is at 42-44ppb, it will be zero at the motorway (like typical on high traffic roads). The further reaction requires dilution with O3 rich air. Is this considered in the CAABA model? How this would effect the result?

l. 178: The estimated ratio from CAABA is 0.7 +/-0.4. It is not clear which solar radiation data are used for this calculation. The same is the case for the steady state conversion factor in l. 188.

Minor points:

Abstract: The whole approach is based on the conversion of NO to NO2, which depends on Ozone. It is important to mention this in the abstract.

l.7: "...concentration over the lowermost 2 to 3km." But what is the most relevant height in such a study. The plume will not be uplifted to several 100m, but will be rather below 100m if you are so close to the highway. So this statement is confusing. What is the expected height of the plume at the "East side" location?

Fig. 1: Following on the plume hight → How the area of highest sensitivity is calculated marked in Fig. 1?

l. 24: Since EURO 6c, the WLTP test cycle is the new test standard. For EURO VI the

WHSC.

l. 30: What do you mean with "need to convert NO to NO2" which sounds like a problem? The measurement systems observe NOx, how the instruments measure internally NOx is not relevant for the derived emission data.

l. 31 & l. 34: The statement "Furthermore, this approach is dependent on the exact position of the emission source and the inlet of the measuring instrument." and "Both approaches are able to resolve the emission of individual vehicles but are depending on the wind field and the position of the exhaust pipe with respect to the measuring instrument." is not correct. Remote sensing and plume chasing observe ratios of gases e.g. NOx/CO2 and derive from this the emission factor. The dilution between emission source and inlet is not relevant for the emission value. It may have an effect if a sufficient signal is captured at all, but not for the value itself. It needs to be correct that these systems directly observe RDE of individual vehicles and measurement position and wind field is not relevant.

l. 33: The statement "However, these require an estimate of the amount of primary NO2 in the exhaust." is only valid for older remote sensing systems, as newer directly measure also NO2 and NO. Additionally the direct NO2 emission is small in relation to NO, thus this error is not so large. An other reason is valid why remote sensing has large errors: The snap shot emission measurement at very specific driving conditions where these systems work are not representative for the average emissions of an individual vehicle and also not necessary on average over many measurements as many driving conditions are not covered (e.g. motorway). That would be the motivation to derive fleet average emission factors and compare it with expected emissions from models.

The advantage of the described method in this manuscript over remote sensing and plume chasing is that it derives the average emission directly, where the other techniques would require a large data set.

l. 57: If the authors state "measure the NO2 emissions of vehicles" this would mean the direct NO2 emissions, not including NO afterwards converted to NO2. But the manuscript focus on NOx emissions, derived from NO2 DSCD.

l. 58: The background NO2 DSCD subtraction is one of the main new methods applied in the manuscript. It should thus also be described in more detail in the method description.

l. 79: The wind measurement was performed upwind, but the important wind speed of the plume is downwind. Can this be different due to shading of trees etc.? What would be the estimated error?

Chapter 2.1: Does not include the measurement date and how meany measurements are performed. It looks like there was only few hours of measurements. What were the conditions during this day? Are they representative. From a statistical point of view this is a quite small data set.

l. 151: The wind speed is only measured at ground level. But the landscape includes trees and hills. The wind speed may not represent the true speed of the plume. Can a better wind speed be estimated from the time shift of "West side" to "East side" of the NO2 data (expecting that these variations are also at plume height)? An additional error for the wind at plume height should be included. What would be the influence if the wind velocity on the motorway (between the trees) is lower? Is the average wind speed derived over all wind speed data points or only in these periods when you have valid DSCD_traffic? This is even more relevant when analyzing data from Fig. A3.

l. 285: Here it is stated that Fig. 2 show CI, but it show NO2 DSCD.

l. 227: Include the applied molar mass (46,01 g·mol−1).

l. 244: The sentence "trucks only account for a small amount of the total traffic volume", is confusing, as they cause a large portion of the total NOx emissions.

Fig. 4: A description is missing why traffic number is only shown for few times. What is

with the gaps?

Fig A1: Include an explanation why CI is different for both instrument even if they point both at 90° at the same location.

A3, Fig A4 and A5: The difference between the two plots is only the averaging of the data over 12min instead of 2s. The explanation why there is no correlation in A4 is hidden in l. 300. As wind speed measurements are not at the location of the plume, the correlation at the high time resolution seem to be prone for errors and confusing. I suggest directly to show only averaged data (A5), where a small time shift has only a minor effect.

References:

Ibrahim, O., Shaiganfar, R., Sinreich, R., Stein, T., Platt, U., and Wagner, T.: Car MAX-DOAS measurements around entire cities: quantification of NOx emissions from the cities of Mannheim and Ludwigshafen (Germany), Atmos. Meas. Tech., 3, 709–721, https://doi.org/10.5194/amt-3-709-2010, 2010.

Galle, B.; Johansson, M.; Rivera, C.; Zhang, Y.; Kihlman, M.; Kern, C.; Lehmann, T.; Platt, U.; Arellano, S.;Hidalgo, S. Network for Observation of Volcanic and Atmospheric Change (NOVAC)—A global network forvolcanic gas monitoring: Network layout and instrument description.J. Geophys. Res.2010,115, D05304

Lashkina and Lashkin: Estimation of nitrogen oxides emissions from petrol and diesel passenger cars by means of on-board monitoring: Effect of vehicle speed, vehicle technology, engine type on emission rates, Transportation Research Part D: Transport and Environment, https://doi.org/10.1016/j.trd.2016.06.008, 2016.

Athanasios, D., Zisimos, T., Stylianos, D., Georgios, T., Anastasios, K., Zissis, S.: Assessment of CO2 and NOx Emissions of One Diesel and One Bi-Fuel Gasoline/CNG Euro 6 Vehicles During Real-World Driving and Laboratory Testing, Frontiers in Mechanical Engineering, 5, 62,
https://www.frontiersin.org/article/10.3389/fmech.2019.00062, 2019.

TNO Report, The Netherlands In-Service Emission Testing Programme for Heavy-Duty 2011-2013, TNO 2014 R10641 / 2, 2014.

---

## Author Comment (AC1) · 10 Sep 2020

**Reply to comments from Referee #1**

We would like to thank the referee for the comprehensive and thoughtful review, and helpful comments which are addressed individually in the response below. The reviewer's comments are included in italics with the responses in blue.

*The study by Lauster et al. describes a new method to quantify the NOx emissions from a motorway using two MAX-DOAS in parallel. This method is new and complementary to the existing ones, the analysis appears valid, and the presentation of the results in the paper is in general clear, although there is room for improvement in this respect. The experiment also addresses a hot topic regarding air quality. This work fits well in the scope of AMT. Therefore this work should be published, once the authors have taken into account the following remarks.*

**Major points:**

*One limitation of this study is its small database. To my understanding, there was only one day of measurements (10 May 2019). This date appears in the main text only in section 3 (I know it is in the caption of Fig. 1). It is fine to demonstrate a new technique with a small database, but this should be clear in the text. That means adding the date of experiment to the sentences of the abstract and conclusion which gives the factor 11+/-7. In case the authors performed more of such measurements but could only use those of May 10 for some reasons, it would be interesting to (briefly) explain what the problems were.*

We have added the date to the respective sentences in abstract and conclusion. Also, we added (l. 39 in the revised manuscript):
"The presented results are based on one day of measurements (10 May 2019) for proof of concepts. Further measurements could then be used to analyse, e.g., different driving conditions in more detail."
Indeed, we only have one measurement day with this setup (including weather station and using this viewing geometry). The primary aim of our study is to present a proof of concept of the measurement method. Further studies could then include different measurement conditions (e.g. weekdays vs. weekend, different seasons) as well as additional measurement sites to investigate different driving conditions (e.g. speed limits, slope of the motorway). Such an extensive study, however, is beyond the scope of this manuscript.

*It's confusing that the legends indicate 'west side', 'east side' in Fig. 2 and Fig. A1, since they show measurements when both instruments were on the west side to record reference measurements. I suggest to label the instruments e.g. A and B across the text and figures instead (keeping the west side, east side where it makes sense).*

We see the point and adapted the legends and the text accordingly.

*l. 125 and below: Can the authors explain why they use the non filtered SCDtraffic estimate in the main text, if they have filtered the clouds in A2? Does the statement that the 'clouds have only a small impact' refer to the 16% of A2? If so, this is more important than the standard error of the mean (5%) and thus not 'a small impact'.*

In the main text, we refer to the unfiltered case as no clear relation between the cloudiness and the $NO_2$ signal is seen. However, we agree to the referee that a more accurate error estimation should include the deviation of 16%. We therefore added this deviation as an additional error to the traffic induced $NO_2$ SCD and the following processing steps. Also, we dropped the sentence that "clouds have a small impact" (l.132) to avoid further confusion. In the end, the additional error has no significant effect on the outcome of the emission estimation.

*l.177: 'Our simulations with CAABA confirm...' -> The O3 concentration is indeed an important parameter is in the NO2/NO evaluation, one can imagine that the atmospheric mixing is as well. Could the authors add a figure with these simulations, e.g in the appendix? If the NO2/NO ratio is stable in the O3 conditions on 10 May 2019 in Mainz, it is interesting to know in which O3 conditions this ratio is not stable.*

We thank the referee for this important comment. Indeed we did not yet consider the titration of ozone close to the source, where NO concentrations are very high and ozone becomes depleted. This will stop further conversion of NO to $NO_2$. However, turbulent mixing with ambient air increases with distance from the source. Thereby, ozone in the emitted air parcel is replenished and the oxidation of NO continues.
In the revised version of the manuscript, we apply a Gaussian dispersion model using Pasquill stability classes (Pandis and Seinfeld, 2006) based on the atmospheric stability on the measurement day. With this dispersion model we estimate the extent of the emission plume and derive the $NO_2$ mixing ratio from our measurements. While turbulence induced by the local topography and obstacles like trees is neglected, it helps to estimate the evolution of $NO_2$ mixing ratio between emission source and measurement location. From the comparison of the dispersion model and the observations, we conclude that the ozone-limited chemical regime only prevails very close to the emission source.
In order to consider this in our emission estimate calculation, we subdivide the transport of the air parcel in two sections: 1) Close to the emission source we assume that only negligible amounts of NO are converted into $NO_2$ and no further conversion takes place as ozone is depleted. 2) Turbulent mixing with ambient air refills the ozone reservoir and NO to $NO_2$ conversion can be described by the CAABA model simulations. For simplicity, we chose the distance at which the initial $NO_2$ mixing ratio of CAABA model simulations is reached as the transition between both sections.
As the new approach shortens the time for NO to $NO_2$ conversion, it is found that the $NO_2/NO_x$ ratio is smaller than assumed in the previous approach without considering ozone limitations. Since both approaches yield the same results within the error estimation, a modification of the given $NO_2/NO_x$ ratio was not deemed necessary.
The revised approach is described in the text (l. 195 and Appendix C1 in the revised manuscript) including a figure to the simulations.

*l. 188 The authors could refer to previous experiments which indicate that it is unlikely that the NO2/NO equilibrium would be reached so close to a source, e.g. the airborne measurements of NOx fluxes from power plants (see for instance the Phd of A. Meier, Uni. Bremen), or similar studies.*

We have added (in l.214 of the revised manuscript):

"However, it is rather unlikely that the equilibrium state is reached so close to the emission source (as also found for airborne measurements of emission fluxes from power plants; Meier, 2018)."
Similar to other studies, the NO/NO$_2$ emission rate only stabilises at a distance of 3-5 km from the source. Therefore, we do not expect to measure the equilibrium state already at a distance of a few hundred metres. But from our simulations we can conclude that a large part of the emitted NO was already converted to NO$_2$.

*In appendix A1, the statement 'As for cloud free condition a constant CI is expected' is misleading since the CI varies, even without any clouds, with the sun position (see e.g. Gielen, 2014). In practice, this statement is only valid because of the short considered time period, please rephrase.*

Agreed and changed to "An almost constant CI is expected for cloud free conditions in this time period."

*The last sentence of the appendix 'a constant wind is advantageous for the measurements' is an important take-home message and should be explicit in Sect 3.2 and in the conclusions.*

Agreed and added in l.173 and l.293 of the revised manuscript.

*I have several smaller suggestions to improve the presentation, see below.*

**Minor points:**

*l.8: 'independent' -> independently ?*

Done.

*l.13: 'A large fraction of the global emissions' -> can the authors be quantitative on this fraction? 'Therefore'-> does not seem an appropriate adverb here since it's not linked by cause to the previous sentence, what about 'Moreover'?*

According to the 5th assessment report of the IPCC (2013), the anthropogenic emissions of NO$_x$ account for approximately three-quarters of the global NO$_x$ emissions. The phrase reads now "About three-quarters of the global emissions of NO$_x$ originate from anthropogenic sources (IPCC, 2013)." In the next sentence, "Therefore" was changed to "Moreover".

*l.30: 'need to convert NO into NO2 as they directly measure the exhaust plume'-> can the author briefly explain why? (the emissions are mainly NO?). It makes sense to detail also since the reference is in German.*

The paragraph was revised, focusing more on the general approaches used in other studies. Thereby, this sentence dropped out. The study by Pöhler and Engel (2019) internally measures NO$_2$, but as they directly measure the exhaust plume mainly NO (which is produced in the combustion process) is present in the sample. Therefore, the sampled NO is converted into NO$_2$ before the measurement.

*I.55: 'the differential SCD yields the integrated tropo concentration of a specific trace gas'. This seems too short to be accurate. Please specify that the integration is along the photon path and that this quantity is relative (differential) to the column in the reference spectrum.*

Agreed and rephrased more detailed. It reads now:
"Then, the differential SCD yields the integrated tropospheric concentration of a specific trace gas along the photon path (for an altitude range from the surface up to about 2 to 3 km; Frieß et al., 2019, and references therein), i.e. the column density relative to the reference spectrum."

*I.68: perpendicular -> almost perpendicular?*

Done.

*l. 71: 'Possible source of NOx' although that may seem obvious to the authors, I suggest to add that 'since no fire was detected in the area' for other readers*

We have added "since no other sources (e.g. fires) were detected in the area".

*Spectral analysis: presenting the DOAS fit parameters (window, cross-sections, polynomial orders...) in a dedicated table would be more readable and synthetic.*

Done.

*l. 98: 'As can be seen in the grey area' -> actually not much can be seen in the grey area due to the y-axis scale of the lower subplot. I suggest to redo this figure 2, with the second subplot zoomed in the time period of the grey area so that we really see that the delta is about 4e14. This would also make the subplots less redundant.*

Thanks for this suggestion. We changed the plot accordingly.

*In the text, the authors should also explain what this delta is in practice (interpolated? one channel assumed constant?) since the measurements do not appear synchronized in time.*

It is correct that the measurements are not ideally synchronised in time. Therefore, to obtain the difference between the two instruments, the time series of one instrument was interpolated onto the time axis of the other. A corresponding sentence was added to the manuscript (l. 106 in the revised manuscript).

*l. 104 'spectra are being integrated' -> '... averaged' ?*

We have changed this to „accumulated".

*l. 285 'as shown in fig.2'-> 'fig. A2'?*

We have changed "90° measurements as shown in Fig. 2" to "90° measurements (compare to Fig. 2)" as we refer to the same 90° measurements for which the $NO_2$ results are depicted in Fig. 2.

*l. 292 'in Fig. A3 where the dashed line indicates ... threshold' -> Fig A2 ?*

Thanks for pointing this out. We mixed up the sentences. It is now corrected to the following:
"The reference was inferred by fitting a 2nd order polynomial to the data and is depicted as dashed line. The filtered time series are displayed in Fig. A3."

*l. 112 It's expected that the error trends follows the RMS, as it is expected that the RMS decreases with increasing integration times. Please add a few words on the physical explanation (shot noise ...)*

We have changed the following sentence
"Although the average RMS decreases for longer integration times, the $NO_2$ retrieval yields the same result regardless of the integration time."
to
"For the short integration times of our measurements, the spectral residual of the fit is dominated by photon shot noise. This is also clearly demonstrated by the observed dependence of the RMS (and the fit error) on integration time. The RMS decreases for longer integration times as the ratio of the photon shot noise to the measured signal increases. In contrast to the fit error decreasing with integration time, the $NO_2$ retrieval yields the same average $NO_2$ DSCDs for different integration times."

*l. 115-116 'Consequently ... to resolve specific traffic event' -> Please break this sentence in two for the sake of readability*

Done.

*l. 160 For the sake of readability, I suggest to be more explicit with the geometric approximation of the AMF at 20° i.e. to write 2.92.*

Agreed and added to the text.

*Figure 4 is important and should be improved. The y axis of panels A and B should be zoomed to better see the variations and mean values. Panels E and F are redundant, the authors could only show one of them (leading to larger remaining subplots and a clearer figure).*

Thanks for this suggestion. We adapted the plot and text accordingly.

*l. 199 'These emission standards' -> 'the emission standards of trucks' (would be clearer for the reader)*

Done.

---

## Author Comment (AC2) · 10 Sep 2020

**Reply to comments from Referee #2**

We would like to thank the referee for the comprehensive and thoughtful review, and helpful comments which are addressed individually in the response below. The reviewer's comments are included in italics font with the responses in blue.

*The manuscript presents a new approach to derive average vehicle NOx emissions on a motorway using passive MAX-DOAS. This method is a further adaptation / modification of emission estimates like performed with car MAX-DOAS [e.g. Ibrahim et al. 2010] or stationary MAX-DOAS for volcanoes [Galle et al. 2010]. While the basic measurement principle is similar, the setup was here adapted to the task of vehicle emission measurements.*

*The applied method is well described and clear. They have the potential to be established for general vehicle emission monitoring of whole fleets. The topic fits well in the scope of AMT. There are two major weak points. First, several parameters which influence the measurement are not well considered. They lead to a further increase of the derived emission factor error, which is already quite large with 4.3+/- 2.5 x 10^19 molec/(ms).*

*Second, the calculations of the expected NOx emissions are incorrect and underestimated, which make the manuscript and likely the measurement principle disputable. This does not mean that a significant higher emission is derived than expected, but a more realistic expected emission would strengthen the value of the manuscript. The manuscript should first be corrected before publication.*

**Major points:**

*Chapter 3.4: Expected traffic emissions*

*The EURO emission standard is a limit which is based on a lab test cycle on a chassis dynamometer (NEFZ, and now WLTP) and only needs to be fulfilled on average over the whole test cycle. The emissions can be for some driving situations higher and for other lower. Especially on a motorway where the engine load is high, emissions are typically higher than the average especially for passenger cars (e.g. HBEFA data base, Lashkina and Lashkin 2016, Athanasios et al. 2019, for trucks e.g. TNO 2016). Second, It is expected and well known, that real driving emissions (RDE) will be higher as the driving and surrounding properties in the test cycle are not realistic (like also, mentioned in the manuscript l. 26.). For trucks this is also limited since EURO VI with a RDE factor of 1.5. This means that EURO VI trucks are allowed to emit on average 1.5\*460mg/kWh = 690mg/kWh in RDE (not the applied 460mg/kWh). For passenger cars the RDE confirmation factor is 2.1 since EURO-6d-Temp. RDE are thus for these diesel cars 2.1\*80mg/km = 168mg/km. For older vehicles RDE is not tested and thus an emission confirmation with a confirmation factor is not defined. In conclusion this means that the emission standard (Table 1) are not the expected RDE even if the vehicles confirm to the emission limit. The error becomes obvious as the expected weighted emission limit of the vehicle fleet (l. 213) is with 116mg/km below the RDE limit newest EURO 6d-temp diesel passenger cars need to confirm (168mg/km). Third, there are engine situations where significant higher emissions are allowed like cold start. If directly comparing measured*

*emissions with calculated emissions, it should be excluded that these driving situations could contribute. Else they need to be considered.*

*There are different ways to handle the comparison more correctly:*
*a) The expected emissions are modelled using the vehicle fleet, number, driving property at the measurement site and emission RDE data from HBEFA data base. This can also be made if it is expected that all vehicles confirm to the legislation or if included known RDE emission values. The expected emissions will increase in comparison to the authors calculation. That does not mean that they are than in agreement with the measurement, but this would allow a comparison between expected and measured average emissions.*
*b) The derived total emission is compared to the average emission limit on the chassis dynamometer (like currently done in the manuscript), however than a direct relation of how much the derived emissions are higher needs to be avoided. It must be clearly stated that the calculated emissions do not represent the expected emissions on the motorway, which is higher even if the vehicles confirm to the legislation. The comparison just gives the reader a relation between the numbers. In general the manuscript would than focus more on the derived total emission and less on the comparison.*
*c) The calculated expected emissions are at least more realistic. That mean that emission factors for motorways need to be used. Additional the RDE conformity factor need to be applied, which however only exist for newer EURO 6 / VI. How to deal with older cars is thus difficult. Additionally some estimated emissions of the trains need to be considered. Even if they are not clearly seen in the DSCD's (like mentioned in l. 130), they are still included in these data.*

First, we thank the referee for this very extensive discussion and ideas to improve the calculation of the expected/theoretical emissions of the vehicle fleet!
As the referee already points out, a more sophisticated assessment using the European emission standards and RDE conformity factors cannot be done in a consistent way as hereto the RDE conformity factors for older emission classes are missing. However, to reduce the risk of confusion we changed „expected emissions" to „theoretical emissions" to emphasise the fact that these values are referring to the European emission standards and not to real driving conditions.
We added a paragraph (l.227 in the revised manuscript):
"The European emission standards are theoretical values for the allowed emissions of different pollutants. They are, however, not the expected emissions under real driving conditions. In order to bring the values in line, so-called Real Driving Emissions (RDE) conformity factors are used for new emission norms (Euro 6; Council of the European Union, 2016). To avoid inconsistencies, in the following only the European emission standards serve to estimate the theoretically expected emissions."

We also added another sentence to Sect. 2.1 Experimental setup:
"The chosen motorway section has a speed limit of 100 km h$^{-1}$. The next access and exit is about 1 km in one direction and 1.5 in the other direction. Acceleration and deceleration should, therefore, only have a minor effect at the measurement site."
The measurement location is thus ideal to measure constant emission, which also encourages investigating the average emission flux over the whole measurement time series.

Additionally, we now analysed the expected emission as given by the HBEFA database. Here, we concentrated on the vehicle categories 'passenger cars' (PC) and 'heavy duty vehicles' (HDV) as these can be readily identified in the camera recordings of the motorway section. It can further be differentiated between hot/cold emission categories. However, cold starts are not to be expected on this motorway section and also they only have little impact on the overall emissions when comparing the values given by the database. We used the aggregated emission factors for $NO_x$ (in units of g/vehkm) of the year 2020. Again comparing the values e.g. to the year 2015, differences especially in the category of HDV can be seen. In total, the effect remains rather small. Using the emission factors of the HBEFA database regarding $NO_x$ emissions, $1.1 \times 10^{19}$ molec/(m s) are to be expected on average. Our measurements show values which are 4+-2 times larger than the calculated emissions. Although the database provides real driving emission factors, there remains a discrepancy to the measurements. Nonetheless, this additional comparison shows that the measurement method yields reasonable results and seems to be able to quantify average emissions of the motorway section. A respective section was added to the revised manuscript (Sect. 3.5).

*l. 209: The number of total travelled distance of trucks may not represent the real truck composition on the motorway. Especially on the motorway typically more foreign trucks are present than on average on the road. A more realistic number can be found from the toll collect system (https://www.bag.bund.de/DE/Navigation/Verkehrsaufgaben/Statistik/Mautstatistik/mautstatistik_node.html).*

We agree to the referee that a considerable amount of non-German trucks has to be expected on the motorway. Analysing the data given by the German toll collect system, however, shows no significant deviation of the distribution with regard to the emission classes although roughly 35% are non-German trucks.

*Chapter 2.2: Deriving DSCD's*

*A spectral fit is missing in the appendix.*

Done.

*l. 110: The given NO2 DSCD error of 0.006 x 10ˆ16 molec cmˆ-2 does not agree to the given RMS values. Please provide the correct NO2 DSCD errors of the spectral analysis. The error for the average DSCD is not reducing with Gaussian error propagation.*

We have replaced "average $NO_2$ error" by "$NO_2$ fit error" and "average RMS" by "RMS" to state more clearly to what we refer. Further, we rephrased "The standard error of the average $NO_2$ DSCD..." to "The standard error of the mean regarding the $NO_2$ DSCD is about $0.006 \times 10^{16}$ molec cm$^{-2}$". The error of $0.006 \times 10^{16}$ molec cm$^{-2}$ here refers to the statistical error of the $NO_2$ DSCD time series, whereas the RMS values (also shown in Fig. 3) are given by the QDOAS analysis and averaged over all data points.

*Also in l. 125 / 126 the given error seems to be calculated with gaussian error propagation of the mean which is not valid. Systematic measurement errors do not behave like a statistical standard error.*

This is correct. We calculated the statistical error in the average traffic induced $NO_2$ SCD using the standard error of the mean. This is a valid approach as systematic errors that impact the calculated difference between the upwind and downwind instrument would also affect the 90° reference spectra. Here, no major deviation between the instruments can be seen (Fig. 2). The fit error of the DSCDs is mainly composed of a measurement noise component and an instrument noise component. Hereby, the instrument noise is largely influenced by the integration time and shows the same trend as discussed above for the RMS. It is concluded that the measurement result is not affected by this. The measurement noise includes statistical fluctuations of the $NO_2$ signal and is thus also not relevant when averaging over longer time spans as done in the retrieval of the averaged, traffic induced $NO_2$ SCD.
A physical explanation is added (l. 117 in the revised manuscript):
"For the short integration times of our instruments, the spectral residual of the fit is dominated by photon shot noise. This is also clearly demonstrated by the observed dependence of the RMS (and the fit error) on integration time. The RMS decreases for longer integration times as the ratio of the photon shot noise to the measured signal increases. In contrast to the fit error decreasing with integration time, the $NO_2$ retrieval yields the same average $NO_2$ DSCDs for different integration times."

*The typical averaging (2s) is very short with resulting noisy NO2 DSCD's. The authors derive an average emission factor over a longer time period. It is explained that there is no significant difference between the different time resolutions (l. 113). It is not clear which difference the authors mean here. The one for the example in Fig. 3? Or the difference for the whole SCD_traffic? Even if the difference is small, I do not understand the argument in l. 116 "to resolve specific traffic events", as none of these events are analyzed in the manuscript. If analyzing averaged spectral data (16 s or longer) the section 2.4 and Fig. 3 can be shorten.*

In l.113 we changed "result" to "average $NO_2$ DSCDs" such that the sentence now reads "...the $NO_2$ retrieval yields the same average $NO_2$ DSCDs for different integration times." The following discussion of differences refers to the deviation of the average $NO_2$ DSCDs for different integration times separately for both instruments, but not to the traffic induced difference between the two instruments.
We stick to the original data (2 s integration time) as we do not expect any information gain/loss when averaging over longer time spans. Although we do not explicitly use the high temporal resolution, the analysis/discussion of the integration time shows that generally it is conceivable to resolve individual emission plumes, e.g. for lower traffic volume (where it might be easier to differentiate single emission plumes) or for higher workloads (at motorway sections that show higher slopes). We added this information to the respective paragraph (l.125 in the revised manuscript).

*The influence of clouds is not clear through the manuscript. Clouds seem to have a large influence on the results. It is not clear if both MAX-DOAS point at 90° at the same location, why a difference in DSCD is observed?*

During the 90° measurements, both instruments were positioned close to each other (less than 2 m distance). Therefore, the spatial mismatch is rather small. Nevertheless, it would be possible that one instrument already sees a cloud edge, whereas the other does not, because of small deviations of the viewing directions. More importantly, the instruments were not synchronised in time (added to l. 106 in the revised manuscript) such that there is a time shift between the measurements of both instruments which induces deviations in the DSCD for changing cloud cover. However, we find a good agreement of the two instruments for cloud-free periods.

*Both should see the same cloud and thus same variation of DSCD. From Fig. 2 it looks like a systematic offset for the "East side" instrument is observed. If such small variations already cause such large difference between the instrument (east side instrument measure higher DSCD), how can you exclude that this is not the case when the instrument measure at different locations at 20°?*

In the revised version of the manuscript, any offset between the two instruments is accounted for as an additional error to the retrieved traffic induced $NO_2$ SCD. Moreover, the effect of clouds is generally smaller for slant viewing directions (20°) compared to the zenith viewing direction. Taking the cloud-free reference spectra assures almost perfect agreement between both instruments. In this case, we do not expect and have no indication of systematic deviations.

*From Fig. 2 only the measurement situation without clouds are used for the reference.*

Yes, to assure that both instruments are evaluated against the same reference conditions.

*The argument in l. 132. that clouds have only a small influence is not clear as Fig. A3 shows the influence also for the emission measurement. With the argument from Fig. 2 (both instrument at west side) also only data without clouds should be used for the emission analysis (like Fig. A3 instead data from Fig. 4).*

In the main text, we refer to the unfiltered case as no clear relation between the cloudiness and the $NO_2$ signal is seen. However, we agree that the statement in l. 132 ("clouds have a small impact") is misleading and was therefore dropped in the revised manuscript. For more accurate error estimation, we added the corresponding deviation of 16%, which might be introduced due to the cloudiness, as an additional error to the traffic induced $NO_2$ SCD. Recalculating the following conversion into the VCD and emission flux, it can be seen that this has no significant effect on the outcome of the emission estimation.

*Are the "West side" and "East side" instrument at the same height? If not, what would be the influence on the NO2 DSCD if they are not at the same height? Could this cause some bias in the DSCD_traffic.*

There is a height difference between the two instruments of about 40 m. However, the light path is in both cases very comparable. Both instruments are set up in the same height above the surface. Therefore, no $NO_2$ molecules go undetected. Moreover, both instruments measure the same background because their viewing

directions are aligned parallel. Small differences in the height (above sea level) are thus negligible.

*Chapter 2.2: Estimation of real driving emissions*

*l. 168: The vehicles emit also directly NO2. The share is for diesel engines (the main NOx emitters) rather high with 30%.*

This is true. We added a statement in l.187 of the revised script ("Especially diesel vehicles also directly emit $NO_2$ (Carslaw et al., 2011, and references therein)."). For a high share of directly emitted $NO_2$, the equilibrium state could be reached closer to the motorway. In any case, the estimate of the equilibrium emission is within the error of the estimation following the simulation results.

*l. 170: Specify that "the share of NO2 in total NOx" need to be known "at the measurement location".*

We have now included this information in the sentence.

*l. 174: The conversion of 2/3 of NO to NO2 is estimated to 4 minutes. This conversion needs O3. As NO emissions are very high at the emissions source, O3 is completely titrated, and thus NO cannot further react to NO2. Even if the background conc. (l. 180) is at 42-44ppb, it will be zero at the motorway (like typical on high traffic roads). The further reaction requires dilution with O3 rich air. Is this considered in the CAABA model? How this would affect the result?*

We thank the referee for this important comment. Indeed we did not yet consider the titration of ozone close to the source, where NO concentrations are very high and ozone becomes depleted. This will stop further conversion of NO to $NO_2$. However, turbulent mixing with ambient air increases with distance from the source. Thereby, ozone in the emitted air parcel is replenished and the oxidation of NO continues.
In the revised version of the manuscript, we apply a Gaussian dispersion model using Pasquill stability classes (Pandis and Seinfeld, 2006) based on the atmospheric stability on the measurement day. With this dispersion model we estimate the extent of the emission plume and derive the $NO_2$ mixing ratio from our measurements. While turbulence induced by the local topography and obstacles like trees is neglected, it helps to estimate the evolution of $NO_2$ mixing ratio between emission source and measurement location. From the comparison of the dispersion model and the observations, we conclude that the ozone-limited chemical regime only prevails very close to the emission source.
In order to consider this in our emission estimate calculation, we subdivide the transport of the air parcel in two sections: 1) Close to the emission source we assume that only negligible amounts of NO are converted into $NO_2$ and no further conversion takes place as ozone is depleted. 2) Turbulent mixing with ambient air refills the ozone reservoir and NO to $NO_2$ conversion can be described by the CAABA model simulations. For simplicity, we chose the distance at which the initial $NO_2$ mixing ratio of CAABA model simulations is reached as the transition between both sections.
As the new approach shortens the time for NO to $NO_2$ conversion, it is found that the $NO_2/NO_x$ ratio is smaller than assumed in the previous approach without considering

ozone limitations. Since both approaches yield the same results within the error estimation, a modification of the given $NO_2/NO_x$ ratio was not deemed necessary. The revised approach is described in the text (l. 195 and Appendix C1 in the revised manuscript) including a figure to the simulations.

*l. 178: The estimated ratio from CAABA is 0.7 +/-0.4. It is not clear which solar radiation data are used for this calculation. The same is the case for the steady state conversion factor in l. 188.*

The CAABA-MECCA simulation takes the location of Mainz to calculate solar radiation at the surface using solar inclination and typical ozone and other gases' distribution in the atmosphere. It takes into account the sun's orbit on our measurement day - without clouds. Although there were scattered clouds present on the measurement day, the photolysis rates for clear sky are roughly appropriate for our measurements. The information was added to the text (l.193 in the revised manuscript).

***Minor points:***

*Abstract: The whole approach is based on the conversion of NO to NO2, which depends on Ozone. It is important to mention this in the abstract.*

We added the following to the abstract:
"Hereto, the ozone-dependent photochemical equilibrium between NO and $NO_2$ is considered."

*l.7: "...concentration over the lowermost 2 to 3km." But what is the most relevant height in such a study. The plume will not be uplifted to several 100m, but will be rather below 100m if you are so close to the highway. So this statement is confusing. What is the expected height of the plume at the "East side" location?*

We have rephrased the sentence as follows:
"One major advantage of the method used here is that MAX-DOAS measurements are very sensitive to the integrated $NO_2$ concentration close to the surface."

*Fig. 1: Following on the plume height! How the area of highest sensitivity is calculated marked in Fig. 1?*

The area of highest sensitivity indicates the area, for which our measurements have the highest sensitivity to the motorway emissions. It is estimated geometrically using the elevation angle of 20° and the estimated plume height of about 200 m. This information is now added to the text (l. 81 in the revised manuscript) and to Fig. 1.

*l. 24: Since EURO 6c, the WLTP test cycle is the new test standard. For EURO VI the WHSC.*

We rephrased the sentence:
"This procedure is standardised depending on the emission class, e.g. by the New European Driving Cycle (NEDC; European Parliament and Council of the European Union, 1970) and since 2017 by the Worldwide harmonized Light vehicles Test

Procedure (WLTP; Council of the European Union, 2017). These include the measurement of exhaust emissions on a chassis dynamometer."
It should be clearer now that different test cycles are used for different emission classes. Important to note is that these test cycles make use of chassis dynamometers. Although the new test cycles are designed for more realistic driving situations, the retrieved values still cannot represent real driving conditions. The same case applies for heavy duty vehicles (World Harmonized Stationary Cycle, WHSC).

*l. 30: What do you mean with "need to convert NO to NO2" which sounds like a problem? The measurement systems observe NOx, how the instruments measure internally NOx is not relevant for the derived emission data.*
*l. 31 & l. 34: The statement "Furthermore, this approach is dependent on the exact position of the emission source and the inlet of the measuring instrument." and "Both approaches are able to resolve the emission of individual vehicles but are depending on the wind field and the position of the exhaust pipe with respect to the measuring instrument." is not correct. Remote sensing and plume chasing observe ratios of gases e.g. NOx/CO2 and derive from this the emission factor. The dilution between emission source and inlet is not relevant for the emission value. It may have an effect if a sufficient signal is captured at all, but not for the value itself. It needs to be correct that these systems directly observe RDE of individual vehicles and measurement position and wind field is not relevant.*
*l. 33: The statement "However, these require an estimate of the amount of primary NO2 in the exhaust." is only valid for older remote sensing systems, as newer directly measure also NO2 and NO. Additionally the direct NO2 emission is small in relation to NO, thus this error is not so large. Another reason is valid why remote sensing has large errors: The snap shot emission measurement at very specific driving conditions where these systems work are not representative for the average emissions of an individual vehicle and also not necessary on average over many measurements as many driving conditions are not covered (e.g. motorway). That would be the motivation to derive fleet average emission factors and compare it with expected emissions from models. The advantage of the described method in this manuscript over remote sensing and plume chasing is that it derives the average emission directly, where the other techniques would require a large data set.*

Thanks for these comments. The paragraph was revised as follows:
"In-situ measurements such as used in vehicle chasing experiments, e.g. performed by Pöhler and Engel (2019), directly measure the exhaust plume of individual vehicles. Others use remote sensing techniques (Carslaw et al., 2011; Chen and Borken-Kleefeld, 2014) to measure exhaust gases across-road. Both approaches are able to resolve the emissions of individual vehicles but it is difficult to derive representative fleet average emission factors, e.g. to compare these with expected emissions from models, as large data sets would be required."
Now the paragraph focuses more on the general approaches that are used for measuring emission factors without manipulating the vehicles (as e.g. for PEMS) – and less on the exact measurement techniques used in the different studies. This should also emphasise the advantage of our method as MAX-DOAS allows to measure the complete vehicle fleet without gaps/undetected emission plumes.

*l. 57: If the authors state "measure the NO2 emissions of vehicles" this would mean the direct NO2 emissions, not including NO afterwards converted to NO2. But the manuscript focus on NOx emissions, derived from NO2 DSCD.*

Agreed and changed to "quantify the $NO_x$ emissions of vehicles".

*l. 58: The background NO2 DSCD subtraction is one of the main new methods applied in the manuscript. It should thus also be described in more detail in the method description.*

Agreed and changed to "Using two MAX-DOAS instruments on the two sides of the motorway allows to measure the background $NO_2$ DSCDs on the upwind side and additionally the traffic induced $NO_2$ on the downwind side. The background $NO_2$ DSCD is then subtracted from the $NO_2$ DSCD on the downwind side and thus yields the $NO_2$ SCD caused by the traffic emissions."

*l. 79: The wind measurement was performed upwind, but the important wind speed of the plume is downwind. Can this be different due to shading of trees etc.? What would be the estimated error?*

It is correct that the measurement of the wind field is a potential source of errors. Nevertheless, we do not expect and also have no indication of systematic differences between the two sides of the motorway. The wind data shows a rather consistent pattern throughout our measurement time series, turbulent processes in the vicinity of the motorway cannot be accounted for in this approach. We added this information to the revised manuscript in l. 166.

*Chapter 2.1: Does not include the measurement date and how many measurements are performed. It looks like there was only few hours of measurements. What were the conditions during this day? Are they representative. From a statistical point of view this is a quite small data set.*

We have added the date explicitly to the sentences that state the factor between measurement and theoretical emissions in the abstract and conclusion of the revised manuscript.
Also, we added (l. 39 in the revised manuscript):
"The presented results are based on one day of measurements (10 May 2019) for proof of concepts. Further measurements could then be used to analyse, e.g., different driving conditions in more detail."
We took only one day of measurements with this setup (including weather station and using this viewing geometry). The primary aim of our study is to present a proof of concept of the measurement method. Further studies could then include different measurement conditions (e.g. weekdays vs. weekend, different seasons) as well as additional measurement sites to investigate different driving conditions (e.g. speed limits, slope of the motorway). Such an extensive study, however, is beyond the scope of this manuscript. We would still rate the conditions during the day representative for that motorway section.
The weather was sunny with broken clouds in the middle and end of the time series. Hereto, compare to Fig. A1-A3. This is also described in l.74 and in the appendix.
A more detailed description of the motorway properties is added to the manuscript (Sect. 2.1).

*l. 151: The wind speed is only measured at ground level. But the landscape includes trees and hills. The wind speed may not represent the true speed of the plume. Can a better wind speed be estimated from the time shift of "West side" to "East side" of the NO2 data (expecting that these variations are also at plume height)? An additional error for the wind at plume height should be included. What would be the influence if the wind velocity on the motorway (between the trees) is lower? Is the average wind speed derived over all wind speed data points or only in these periods when you have valid DSCD_traffic? This is even more relevant when analyzing data from Fig. A3.*

We added (l.166 in the revised manuscript):
"Effects such as turbulence, especially in the vicinity of the motorway, and changing wind fields at plume height lead to uncertainties which can, however, not be readily quantified."
There are two effects influencing the wind velocity. On the one hand, turbulence especially close to the motorway induces mixing which cannot be assessed easily. However, turbulent effects should statistically cancel out over longer time periods.
On the other hand, the wind velocity increases with height and therefore the measured $NO_2$ signal would be underestimated. Moreover, less NO would be converted to $NO_2$ as the air parcel moves faster to the downwind measurement site – again leading to an underestimation of the retrieved emission. But since the plume is confined within the lowest 200 m, this effect should be quite small.
Also, applying the cloud filter to the wind data, i.e. filtering out the same time periods as indicated by the colour index of the DOAS data, shows no significant difference (average wind velocity without cloud filter: 2.8+-1.0 m/s;
 average wind velocity with cloud filter: 2.9+-1.0 m/s).

*l. 285: Here it is stated that Fig. 2 shows CI, but it shows NO2 DSCD.*

We have changed "90° measurements as shown in Fig. 2" to "90° measurements (compare to Fig. 2)" as we refer to the same 90° measurements for which the $NO_2$ results are depicted in Fig. 2.

*l. 227: Include the applied molar mass (46,01 g/mol).*

Done.

*l. 244: The sentence "trucks only account for a small amount of the total traffic volume", is confusing, as they cause a large portion of the total NOx emissions.*

We changed it to "trucks only account for parts of the total traffic volume".

*Fig. 4: A description is missing why traffic number is only shown for few times. What is with the gaps?*

As stated in l.133/134, the amount of traffic was only counted over one-minute intervals on a sample basis. The bars in Fig. 4 depict the number of cars and trucks for the respective points in time.

*Fig A1: Include an explanation why CI is different for both instrument even if they point both at 90° at the same location.*

We added to the text (l. 331 in the revised manuscript):
„The offset of the CI between the two instruments can be ascribed to the specific instrumental properties as the instruments are not absolutely radiometrically calibrated."
Internal properties explain the deviation of the absolute values in CI for the two instruments. However, for the analysis the respective instrument-specific reference spectrum is taken. Therefore, the outcome is not affected by different radiometric characteristics.

*A3, Fig A4 and A5: The difference between the two plots is only the averaging of the data over 12min instead of 2s. The explanation why there is no correlation in A4 is hidden in l. 300. As wind speed measurements are not at the location of the plume, the correlation at the high time resolution seem to be prone for errors and confusing. I suggest directly to show only averaged data (A5), where a small time shift has only a minor effect.*

We agree to the referee and removed Fig. A4 from the manuscript. The text was adapted accordingly.

---

## Author Response (AR2)

Dear Editor,

We are happy to submit an updated version of the manuscript 'Estimating real driving emissions from MAX-DOAS measurements at the A60 motorway near Mainz, Germany' (amt-2020-125).

The final remarks raised by the referee have been addressed individually and are attached to this response. We also provide a marked-up version of the manuscript in order to expose the modifications we made.

The main difference is the adaptation of Fig. 3 which now additionally includes a panel showing the $NO_2$ fit error. It was also extended to longer integration times showing that the $NO_2$ emission retrieval is not influenced by this.

Kind regards,

Bianca Lauster

Attachments:
- Authors' response to the comments by Referee #2
- Revised version of the manuscript with tracked-changes

**Reply to comments from Referee #2**

We would like to thank the referee for the thorough feedback and corrections which are addressed individually in the response below. The reviewer's comments are included in italics with the responses in blue.

*The authors have answered all questions and performed according corrections. There are some remaining points.*

*There are corrections made in the according sections, but they were not included in the abstract or other sections:*

*1. Line 6: "the measured emissions exceed the maximum expected emissions calculated from the European emission standards by a factor of …" - here it must state "emissions exceed the maximum emissions calculated from the European emission standard for lab test cycle by a factor of…". The theoretical expected emission value would be the HBEFA calculated emission value. Same need to be corrected in line 256.*

Thank you for pointing out these inconsistencies. We changed the corresponding passages according to your suggestions.

*2. Line 22: Here it also should be noted (like in chapter 3.4) that the description is for light duty vehicles.*

Thanks again. We have corrected this in the revised manuscript.

*Added information needs correction:*

*3. L. 229: RDE conformity is applied since "EURO 6d-temp", not since "EURO 6".*

Done.

*4. The authors included a HBEFA emission (chapter 3.5) which gives a good theoretical expected emission value. It would be helpful for the reader if the HBEFA calculation is described in the appendix.*

We have added a respective section in the appendix (C2) giving the values taken from the HBEFA database and describing in more detail the underlying calculation.

*Some points still remain unclear:*

*5. Chapter 2.3: The authors added information on the error estimation. From comparing the given error in line 105, with Fig. 2, it is still unclear how the deviation from the plot with a range from 0.9 to 1.7x10^14 result in the error of 0.4x10^14 molec cm-2.*

As the referee pointed out, the offset between the two instruments ranges from 0.9 to 1.7 x $10^{14}$ molec cm$^{-2}$ and is thus larger than the included error of 0.4 x $10^{14}$ molec cm$^{-2}$. However, this offset is present in all spectra, i.e. for the 90° reference spectra as well as during the measurement time series at 20° elevation angle. Thus,

choosing the same reference period assures that both instruments are analysed against the same (atmospheric) background conditions. The offset is thereby compensated for in the spectral analysis of each measurement time series.

The decisive factor for estimating the error is then not the difference between the two instruments as depicted in the lower panel of Fig. 2 but that this difference is constant with time. Thus, the standard deviation, as stated in the text ($0.4 \times 10^{14}$ molec cm$^{-2}$), yields a reasonable error estimate. This value is also not critical to the further results. The difference itself, i.e. the offset between the two instruments, is not impacting the measurement outcome.

*6. Chapter 3.1: The authors show that the spectral NO2 fit is mainly shot noise dominated. But this does not exclude that there are systematic errors included. For the calculation of the final error the authors apply shot noise standard error propagation. But the authors still did not demonstrate that also the same results and same errors are achieved if first the spectra are averaged to achieve lower NO2 DSCD errors. The high temporal resolution of the NO2 measurement is not needed for the data analysis but lower NO2 DSCD errors would be more convincing.*

We thank the referee for this important follow-up question. We have now added to Fig. 3 the result for longer integration times as well as a panel showing the NO$_2$ fit error. We also analysed again in more detail any systematic features that appear in the RMS of the spectral analysis. It is found that systematic structures become visible for longer integration times, for which the noise levels decrease (as expected). For one of both instruments, we find that the amplitude of the systematic features is of similar magnitude as the amplitude of the shot noise. At the moment, we do not exactly know what causes these systematic structures. However, from detailed analyses we found that they are almost identical in the residuals for all spectra and there is no systematic evolution with time. This is also consistent with the fact that there is no change in the outcome of the measurement time series regardless of the integration time. Furthermore, the comparison of the reference period for the two instruments showed no influence due to these structures. Therefore, we conclude that these systematic errors are unproblematic for our setup and analysis procedure. We adapted the corresponding passage and clarified the statement about the importance of shot noise and possible systematic errors (Sect. 2.3, especially ll.123 to 126).

*7. What is the NO2 DSCD error from the fit in Figure A1? The fit is very noisy and not convincing to derive a small DSCD difference from two of these noisy fits. How does a fit look like of an averaged spectrum over a longer time?*

We have added the NO$_2$ fit error in the figure caption. It amounts $0.53 \times 10^{15}$ molec cm$^{-2}$ which is about 13% of the respective slant column density. The investigation of different integration times (Sect. 2.4) and the comparison of the reference spectra (Sect. 2.3) confirm that the result is robust. It can be concluded that the NO$_2$ fit error has no significant impact on the emission retrieval. Please see hereto also the answer and changes resulting from the previous comment.

[revised manuscript text omitted]